# Interpreting CLIP with Sparse Linear Concept Embeddings (SpLiCE 🔗)

**Usha Bhalla**[*]
Harvard University [a,b]

**Alex Oesterling**[*]
Harvard University [b]

**Suraj Srinivas**
Harvard University [b]

**Flavio P. Calmon**[†]
Harvard University [b]

**Himabindu Lakkaraju**[†]
Harvard University [b,c]

## Abstract

CLIP embeddings have demonstrated remarkable performance across a wide range of multimodal applications. However, these high-dimensional, dense vector representations are not easily interpretable, limiting our understanding of the rich structure of CLIP and its use in downstream applications that require transparency. In this work, we show that the semantic structure of CLIP's latent space can be leveraged to provide interpretability, allowing for the decomposition of representations into semantic concepts. We formulate this problem as one of sparse recovery and propose a novel method, Sparse Linear Concept Embeddings (SpLiCE 🔗), for transforming CLIP representations into sparse linear combinations of human-interpretable concepts. Distinct from previous work, SpLiCE is task-agnostic and can be used, without training, to explain and even replace traditional dense CLIP representations, maintaining high downstream performance while significantly improving their interpretability. We also demonstrate significant use cases of SpLiCE representations including detecting spurious correlations and model editing. Code is provided at `https://github.com/AI4LIFE-GROUP/SpLiCE`.

## 1 Introduction

Natural images include complex semantic information, such as the objects they contain, the scenes they depict, the actions being performed, and the relationships between them. Machine learning models trained on visual data aim to encode this semantic information in their representations to perform a wide variety of downstream tasks, such as object classification, scene recognition, segmentation, or action prediction. However, it is often difficult to enforce explicit encoding of these semantics within model representations, and it is even harder to interpret these semantics post hoc to better understand what models may have learnt and how they leverage this information. Further, model representations can be brittle, encoding idiosyncratic patterns specific to individual datasets and modalities instead of general human-interpretable semantic information. Multimodal models have been proposed as a potential solution to this issue, and methods such as CLIP [1] have empirically been found to provide highly performant, semantically rich representations of image data. The richness of these representations is evident from their high performance on a variety of tasks, such as zero-shot classification and image retrieval [1], image captioning [2], and image generation [3]. However, despite their performance, it remains unclear how to quantify the semantic content contained in their dense representations. In this work, we answer the question: *can we decompose*

---

[*] Equal contribution, order by coin flip.    [†] Equal contribution, alphabetical order.
[a] Kempner Institute for the Study of Natural & Artificial Intelligence
[b] School of Engineering and Applied Sciences    [c] Harvard Business School

38th Conference on Neural Information Processing Systems (NeurIPS 2024).

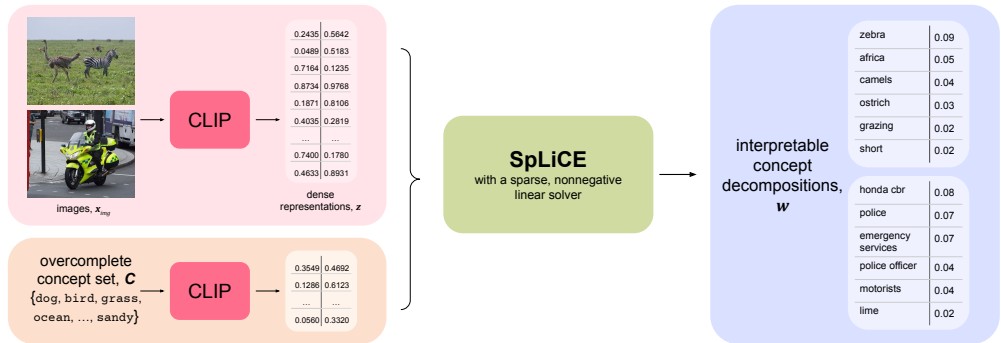

Figure 1: Visualization of SpLiCE, which converts dense, uninterpretable CLIP representations ($\mathbf{z}$) into sparse semantic decompositions ($\mathbf{w}$) by solving for a sparse nonnegative linear combination over an overcomplete concept set ($\mathbf{C}$).

*CLIP embeddings into human-interpretable representations of the semantic concepts they encode?* This can provide insight into the types of tasks CLIP can solve, the biases it may contain, and the manner through which downstream predictions are made.

Existing literature in areas such as concept bottleneck models [4], disentangled representation learning [5], and mechanistic intepretability [6] have proposed various approaches to understanding the semantics encoded by representations. However, these methods generally require predefined sets of concepts [7], data with concept labels [8], or rely on qualitative visualizations, which can be unreliable [9]. Similar to these lines of work, we aim to recover representations that reflect the underlying semantics of the inputs. However, distinct from these works, we propose to do this in a task-agnostic manner and without concept datasets, training, or qualitative analysis of visualizations.

Our method, SpLiCE, leverages the highly structured and multimodal nature of CLIP embeddings for interpretability, and decomposes CLIP representations via a semantic basis to yield a sparse, human-interpretable representation. Remarkably, these interpretable SpLiCE embeddings have favorable accuracy-interpretability tradeoffs when compared to black-box CLIP representations on metrics such as zero-shot accuracy. Our overall contributions are:

- In Sections 3 and 4, we formalize the sufficient conditions under sparse decomposition of CLIP is feasible, and introduce SpLiCE, a novel method that decomposes dense CLIP embeddings into sparse, human-interpretable concept embeddings.

- Our extensive experiments in Section 5 reveal that SpLiCE recovers highly sparse[1], interpretable representations with high performance on downstream tasks, while accurately capturing the semantics of the underlying inputs.

- In Section 6, we present two case studies for applying SpLiCE: spurious correlation detection, and model editing. Using SpLiCE, we uncover a spurious correlation in the CIFAR100 dataset, where we find the "woman" concept and the "swimwear" concept to be correlated owing to the prevalence of women in swimwear in CIFAR100.

## 2 Related Work

**Linear Representation Hypothesis.** In language modeling, the *linear representation hypothesis* suggests that many semantic concepts are approximately linear functions of model representations [10, 11, 12, 13, 14]. Recent work has also shown that multimodal models encode concepts additively, behaving like bags-of-words representations [15]. Relatedly, [16, 17] show that there exists a linear mapping between image and text embeddings in arbitrary models. Our work makes use of these distinct but related observations to convert dense CLIP representations to sparse semantic ones.

**Concept Bottlenecks and Attribute Learning.** Concept Bottleneck Models (CBMs) [18], and attribute-based models [19, 20, 21] learn intermediate representations of scores over concepts or image

---

[1]we recommend and use sparsity levels of $\sim$ 10-30 in practice

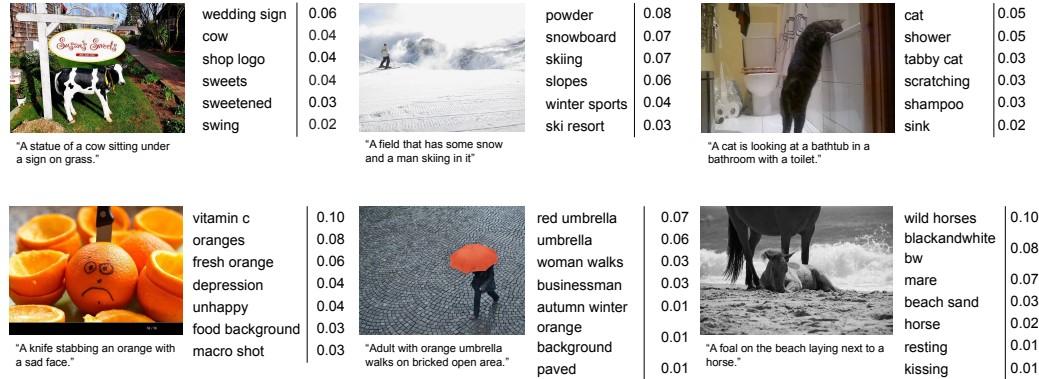

| | | | | | | | | |
|---|---|---|---|---|---|---|---|---|
| wedding sign | 0.06 | | powder | 0.08 | | cat | 0.05 | |
| cow | 0.04 | | snowboard | 0.07 | | shower | 0.05 | |
| shop logo | 0.04 | | skiing | 0.07 | | tabby cat | 0.03 | |
| sweets | 0.04 | | slopes | 0.06 | | scratching | 0.03 | |
| sweetened | 0.03 | | winter sports | 0.04 | | shampoo | 0.03 | |
| swing | 0.02 | | ski resort | 0.03 | | sink | 0.02 | |

"A statue of a cow sitting under a sign on grass."     "A field that has some snow and a man skiing in it"     "A cat is looking at a bathtub in a bathroom with a toilet."

| | | | | | | | | |
|---|---|---|---|---|---|---|---|---|
| vitamin c | 0.10 | | red umbrella | 0.07 | | wild horses | 0.10 | |
| oranges | 0.08 | | umbrella | 0.06 | | blackandwhite bw | 0.08 | |
| fresh orange | 0.06 | | woman walks | 0.03 | | mare | 0.07 | |
| depression | 0.04 | | businessman | 0.03 | | beach sand | 0.03 | |
| unhappy | 0.04 | | autumn winter | 0.01 | | horse | 0.02 | |
| food background | 0.03 | | orange background | 0.01 | | resting | 0.01 | |
| macro shot | 0.03 | | paved | 0.01 | | kissing | 0.01 | |

"A knife stabbing an orange with a sad face."     "Adult with orange umbrella walks on bricked open area."     "A foal on the beach laying next to a horse."

Figure 2: Example images from MSCOCO shown with their captions below and their concept decompositions on the right. We display the top seven concepts for visualization purposes, but images in the figure had decompositions with 7-20 concepts.

attributes for use with a final linear classification head, creating interpretable concept representations. However, these require expert-labeled concept or attribute datasets to train, which is expensive. Recent work on concept-bottlenecks for multimodal models avoids needing such labeled datasets, but still requires concept labels for specific tasks, which is obtained by querying large language models (LLMs) [22, 23, 24], making these methods task-specific and heavily reliant on the domain knowledge and subject to the biases of LLMs. On the other hand, SpLiCE uses a large-scale and overcomplete concept dictionary, avoiding dependence on training, specific domain knowledge, or a downstream task. Consequently, it can even be applied to understand unstructured, unsupervised image datasets in a label-free manner.

**Mechanistic Interpretability and Disentanglement.** Mechanistic interpretability explains representations through model activations, by labeling circuits and neurons in networks with feature visualization [6, 25] or by measuring concept activations and directions in latent space [26, 27, 7, 28, 29, 30]. Recent work [31] combines these methods, using dictionary learning to extract visual concept activations, whose semantics can be identified via feature visualization. Work in disentangled representation learning has developed architectures that capture independent factors of variation in data [8, 32, 33, 5, 34, 35], allowing for manual probing of disentangled representations for human-interpretable concepts. In both mechanistic interpretability and disentangled representation learning, methods typically rely on labeled concept sets, manual labeling of visualizations, or computationally intensive searches over data and latent representations or neurons to identify concepts. However, associating human-interpretable semantics with arbitrary neurons or latent directions is challenging, leading to the unreliability [9, 36] exhibited by such methods. Our approach side-steps this issue by decomposing CLIP representations into a predetermined set of concepts.

**CLIP Interpretability.** Many recent works leverage the semantic structure of CLIP and its text encoder to interpret representations. For example, [37], [38], and [39] construct concept similarity scores of image embeddings for use by downstream CBMs or probes, but these representations are not interpretable due to their lack of sparsity and the presence of negative concepts. Chen et al. [40] create a custom vision-language architecture with a sparse latent dictionary, but it requires training from scratch and cannot be used post-hoc to explain existing models. Gandelsman et al. [41] also leverage the text encoder of CLIP to explain components of the image embedding, but are limited to ViT architectures and take a mechanistic interpretability-style approach requiring a labeled text dataset. Chattopadhyay et al. [22] build concept bottlenecks for specific classification tasks by expressing CLIP image representations as a sparse linear combination of task-specific concept vectors. However, their decomposition includes negative concepts, reducing interpretability, and uses task-specific concept dictionaries. Grootendorst [42] generate textual topics of datasets through multimodal topic modeling, which cannot provide explanations of individual representations. Distinct from these works, SpLiCE is more interpretable due to its sparsity, overcompleteness, and non-negativity, and is task-agnostic, aiming to serve as a drop-in replacement for black-box CLIP representations without requiring training.

Table 1: Sanity checking the linearity of CLIP Embeddings.

|  | $w_a$ | $w_b$ | $\text{COSINE}(\hat{z}, z)$ |
|---|---|---|---|
| IMAGENET | $0.48 \pm 0.09$ | $0.45 \pm 0.09$ | $0.76 \pm 0.05$ |
| CIFAR100 | $0.45 \pm 0.08$ | $0.42 \pm 0.08$ | $0.75 \pm 0.03$ |
| MIT STATES | $0.48 \pm 0.09$ | $0.45 \pm 0.09$ | $0.76 \pm 0.05$ |
| COCO TEXT | $0.59 \pm 0.12$ | $0.47 \pm 0.12$ | $0.88 \pm 0.04$ |

## 3  When do Sparse Decompositions Exist?

In this section, we aim to answer the question: *under what conditions can CLIP representations be decomposed into sparse semantic representations*? To do so, we must reason about both the properties of CLIP as well as the properties of the underlying data.

**Notation.** Let $\mathbf{x}^{\text{img}} \in \mathbb{R}^{d_i}$, $\mathbf{x}^{\text{txt}} \in \mathbb{R}^{d_t}$ be image and text data, respectively. Given the CLIP image encoder $f : \mathbb{R}^{d_i} \to \mathbb{R}^d$ and text encoder $g : \mathbb{R}^{d_t} \to \mathbb{R}^d$, we define CLIP representations in $\mathbb{R}^d$ as $\mathbf{z}^{\text{img}} = f(\mathbf{x}^{\text{img}})$ and $\mathbf{z}^{\text{txt}} = g(\mathbf{x}^{\text{txt}})$. Our method uses dictionary learning to approximate $\mathbf{z}^{\text{img}}$ with a concept decomposition $\mathbf{w}^* \in \mathbb{R}_+^c$ over a fixed concept vocabulary $\mathbf{C} \in \mathbb{R}^{d \times c}$. We define the resulting reconstruction of $\mathbf{z}^{\text{img}}$ from $\mathbf{C}$ and $\mathbf{w}^*$ as $\hat{\mathbf{z}}^{\text{img}}$.

The goal of our method is to approximate $f(\mathbf{x}^{\text{img}}) \approx \mathbf{C}\mathbf{w}^*$, such that $\mathbf{w}^*$ is non-negative and sparse, and in this section we formalize when this is possible. We begin by considering a data-generating process for coupled image and text samples. Specifically, we model the generative process parameterized by a $k$-dimensional latent concept vector $\omega \in \mathbb{R}_+^k$ and a random noise vector $\epsilon \in \mathbb{R}^l$ as

$$\mathbf{x}^{\text{img}} = h^{\text{img}}(\omega, \epsilon), \quad \mathbf{x}^{\text{txt}} = h^{\text{txt}}(\omega, \epsilon), \quad \omega \sim \rho, \quad \epsilon \sim \phi,$$

where $\rho$ is a prior distribution over semantic concepts, $\phi$ is a prior distribution over nonsemantic concepts (such as camera orientation and lighting for images or arbitrary choices between synonyms for text), and $h^{\text{img}} : \mathbb{R}^{k+l} \to \mathbb{R}^{d_i}$, and $h^{\text{txt}} : \mathbb{R}^{k+l} \to \mathbb{R}^{d_t}$ represent the real-world data-generating process from latent variables $(\omega, \epsilon)$ to images and text respectively. Here, each coordinate $\omega_i \in \mathbb{R}_+$ encodes the degree of prevalence of the $i^{\text{th}}$ concept in the underlying data. We now list a set of sufficient conditions for our data-generating process and CLIP that admit a sparse decomposition of images into concepts.

**Sufficient Conditions for Sparse Decomposition.**

1. Images and text are sparse in concept space: for some $\alpha \ll k$, we have $\|\omega\|_0 \leq \alpha, \forall \omega \sim \rho$.
2. CLIP captures semantic concepts $\omega$ and not $\epsilon$: $\forall \epsilon, \epsilon', f \circ h^{\text{img}}(\omega, \epsilon) = f \circ h^{\text{img}}(\omega, \epsilon')$ and similarly for $h^{\text{txt}}$.
3. CLIP is linear in concept space: $g \circ h^{\text{txt}}$ and $f \circ h^{\text{img}}$ are linear in $\omega$.
4. CLIP image and text encoders are aligned: for a given $\omega$, $f \circ h^{\text{img}}(\omega, \epsilon) = g \circ h^{\text{txt}}(\omega, \epsilon)$.

We emphasize that the goal of enumerating a set of sufficient conditions for sparse decomposition is not to claim that these exactly hold in practice, but rather to reason about when sparse decompositions–as done in this work–are appropriate. In the Appendix (Section A.1, Prop. 1) we formalize and prove this claim, but in the interest of simplicity we keep the discussion here informal. We note that many of these are natural; Assumption 1 reflects how real-world images and text are simple and rarely contain complex semantic content, and the CLIP training process optimizes for Assumption 2 and $4^2$. Of these, the most critical one is Assumption 3, which closely relates to the linear representation hypothesis [11], which we investigate below.

**Sanity Checking CLIP's Linearity.**  We provide evidence for the third assumption, the linearity of CLIP, in a toy setting. We begin by asking the following question to confirm the general linearity of CLIP embeddings: "if two inputs are concatenated, does their joint embedding equal the average

---

[2]In practice we find that CLIP's image and text encoders are not fully aligned, so we apply a preprocessing step (Sec 4.1).

of their two individual embeddings?". For the image domain, we combine two images, $x_a, x_b$, to form their composition $x_{ab}$ by placing $x_a$ in the top left quarter and $x_b$ in the bottom right quarter of a blank image. For the text domain, we simply append text $x_b$ to text $x_a$ to form $x_{ab}$. We then embed $x_a, x_b, x_{ab}$ with CLIP to get $z_a, z_b, z_{ab}$. Solving the equation $w_a * z_a + w_b * z_b = z_{ab}$ for scalar weights $w_a, w_b$ then allows us to assess the linearity of $z_a, z_b, z_{ab}$. We report $w_a, w_b$ and the cosine similarity between $\hat{z}_{ab} = [z_a, z_b] \cdot [w_a, w_b]$ and $z_{ab}$ in Table 1.

In general, we find that the composition of two inputs results in an embedding that is approximately equal to the average of the two input components, with $w_a, w_b$ being very close to 0.5 across all datasets and for both modalities, providing preliminary evidence for the linearity of CLIP embeddings for both image and language.

# 4 Method

In this section, we introduce SpLiCE, a method for expressing CLIP's image representations as sparse, nonnegative, linear combinations of concept dictionary elements. We begin by framing this problem as one of sparse recovery. We then discuss our design choices, including how we choose the concept dictionary and how to address the modality gap between CLIP's images and text representations. Finally, we formalize the optimization problem used in this work.

## 4.1 Sparse Nonnegative Concept Decomposition

Our goal is to construct decompositions of dense CLIP representations that are human-interpretable, useful, and faithful. To do so, we formulate decomposition as a sparse recovery problem with three main desiderata. First, for the decompositions to be interpretable to humans they must be comprised of human interpretable atoms. We argue that language is a naturally interpretable interface for humans, and construct our concept vocabulary **C** out of 1- and 2-word atoms, such as "coffee", "silver", and "birthday party". Second, our decompositions must be simple and concise, which can be formulated as a sparsity constraint on the recovery. A large body of work in computational linguistics [43, 44, 45, 14], neuroscience [46, 47], and interpretability [48, 49, 30] have demonstrated that a human-aligned semantic model should be sparse in representation. Furthermore, [48] found that users can best understand explanations with fewer than 32 concepts while in linguistics, [50, 51, 52] find participants describe concepts and objects with up to 20 semantic properties, motivating our desiderata of sparsity. Third, our decompositions must be constructive, i.e., we must decompose representations in terms of their constituent concepts. For this reason, we require the weights of decompositions to be strictly nonnegative, to avoid having "negative" concept weights which do not always carry semantic meaning. Furthermore, prior work by Zhou et al. [30] has argued that *negations of concepts are not as interpretable as positive concepts.* More specifically, while a small set of concepts have well-defined antonyms which may be viewed as their negative counterparts (``day'' $\leftrightarrow$ ``night''), negative concepts do not carry semantic meaning in general (``tiger'' $\leftrightarrow$ ??). Furthermore, we find that even when antonyms exist, they are not negatives of each other in CLIP latent space (see Appendix B.10). To avoid dependence on negative weights and ensure that all concepts are captured, we construct an overcomplete dictionary containing a wide range of concepts, including antonyms. We build on top of this literature and provide a semantic decomposition satisfying these properties suitable for multimodal models like CLIP.

**Concept Vocabulary.** Natural language is an intuitive, interpretable, and compact medium for communicating semantic information. Thus, we choose to represent the semantic content contained in CLIP embeddings as combinations of natural language semantic concepts, where we define concepts as semantic units that can be expressed concisely, by one- or two-word phrases. Given that CLIP is used in a wide variety of downstream applications and is trained without a specific task in mind, we want our concept dictionary to be task-agnostic and to span *all possible concepts CLIP could have learnt*. To construct this vocabulary, we consider the most frequent one- and two-word bigrams in the text captions of the LAION-400m dataset [53], the dataset that most CLIP variants are trained on. We filter the captions to remove any NSFW samples and prune our concept set such that no two concept embeddings have a cosine similarity greater than 0.9. We also remove bigrams highly similar ($> 0.9$ cosine similarity) to the average of their individual words. We finally choose the top $10,000$ most common single-word concepts and the top $5000$ most common two-word concepts as our concept vocabulary. We note that this vocabulary offers distinct advantages over those used in prior works.

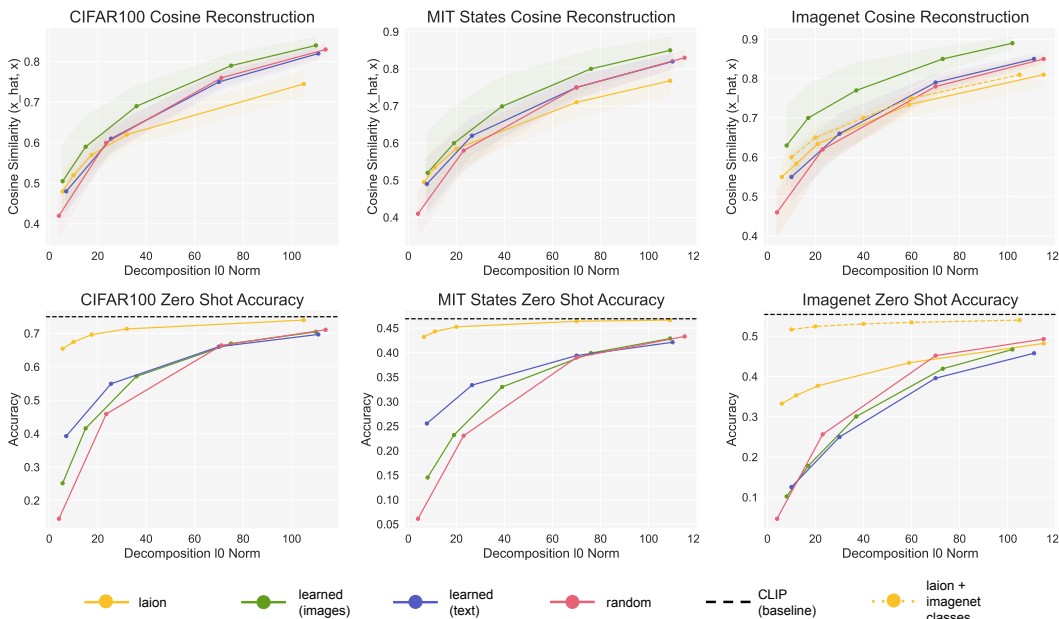

Figure 3: Performance of SpLiCE decomposition representations on zero-shot classification tasks (bottom row) and cosine similarity between CLIP embeddings and SpLiCE embeddings (top row). Our proposed semantic dictionary (yellow) closely approximates CLIP on zero-shot classification accuracy, but not on the cosine similarity. This indicates that SpLiCE captures the semantic information in CLIP, but not its non-semantic components, explaining both the high zero-shot accuracy and low cosine similarity. See §5.2 for discussion.

In particular, it is *task-agnostic*, meaning that the efficacy of the decomposition is (in principle) independent of individual datasets. Furthermore, this dataset imposes minimal priors from outside curators, such as human experts or LLMs [22, 23, 24]. This allows us to interpret data through the lens of CLIP, to understand the information encoded, including potential biases and mistakes.

**Modality Alignment.** In order to decompose images into nonnegative combinations of text concepts, we must ensure that our concept set spans the space of possible image embeddings. However, [54] show the existence of a modality gap in CLIP, where image and text embeddings can lie in non-identical spaces on the unit sphere. We empirically find that CLIP image and text embeddings exist on two cones, as the distribution of pairwise cosine similarities between pairs of MSCOCO images and pairs of MSCOCO text captions concentrate at positive values, whereas the distribution of pairwise cosine similarities across modalities concentrates closer to zero. (See Appendix Fig. 7). Not only does this prevent nonnegative decomposition, it also violates Assumption 4 from Section 3. To rectify this, we mean-center CLIP images with the image cone mean, estimated over MSCOCO ($\mu_{\mathbf{img}}$), and compute decompositions over the mean-centered concept vocabulary ($\mu_{\mathbf{con}}$). Note that the embeddings need to be re-normalized after centering to ensure they lie on the unit sphere. To convert our decompositions back into dense representations ($\hat{\mathbf{z}}^{\text{img}}$), we uncenter the normalized dense embeddings $\hat{\mathbf{z}}^{\text{img}}$ by adding the image mean back in and normalizing once again, to ensure they lie on the same cone as the original CLIP embeddings ($\mathbf{z}^{\text{img}}$).

**Optimization Problem.** Our optimization problem is formulated as follows. Let $\sigma(\mathbf{x}) = \mathbf{x}/\|\mathbf{x}\|_2$ be the normalization operation. Given a set of semantic concepts $\mathbf{x}^{\text{con}} = [\text{``dog''}, \text{``tabby cat''}, \text{``cloudy''}, \cdots ]$, we construct a centered vocabulary $\mathbf{C} = [\sigma(g(\mathbf{x}_1^{\text{con}}) - \mu_{\text{con}}), \cdots, \sigma(g(\mathbf{x}_c^{\text{con}}) - \mu_{\text{con}})]$, where we recall that $g(\cdot)$ is the CLIP text encoder. Now, given the dictionary $\mathbf{C}$ and a centered CLIP embedding $\mathbf{z} = \sigma(\mathbf{z}^{\text{img}} - \mu_{\text{img}})$, we seek to find the sparsest solution that gives us a cosine similarity score of at least $1 - \epsilon$ for some small $\epsilon$:

$$\min_{\mathbf{w} \in \mathbb{R}_+^c} \|\mathbf{w}\|_0 \text{ s.t. } \langle \mathbf{z}, \sigma(\mathbf{C}\mathbf{w}) \rangle \geq 1 - \epsilon. \tag{1}$$

As is standard practice, we relax the $\ell_0$ constraint and reformulate this as a minimization of MSE with an $\ell_1$ penalty, to construct the following convex relaxation[3] of Eq. (1):

$$\min_{\mathbf{w} \in \mathbb{R}_+^c} \|\mathbf{C}\mathbf{w} - \mathbf{z}\|_2^2 + 2\lambda \|\mathbf{w}\|_1. \qquad (2)$$

Given the solution to the above problem $\mathbf{w}^*$, our reconstructed embedding is: $\hat{\mathbf{z}}^{\text{img}} = \sigma(\mathbf{C}\mathbf{w}^* + \mu_{\text{img}})$.

## 5 Experiments

In this section, we evaluate our method to ensure that SpLiCE decompositions are interpretable, performant, and accurately reflect the semantic content of representations.

### 5.1 Setup

**Models.** All experiments shown in the main paper are done with the OpenCLIP ViT-B/32 model [55] with results for an additional model in Appendix B.14. For all zero-shot classification tasks, we use the prompt template "A photo of a {}". **Datasets.** We use CIFAR100 [56], MIT States [57], CelebA [58], MSCOCO [59], and ImageNetVal [60] for our experiments with results for additional datasets in the Appendix (Section B.4)

**Decomposition.** For all experiments involving concept decomposition, we use sklearn's [61] Lasso solver with a non-negativity flag and an $l_1$ penalty that results in solutions with $l_0$ norms of 5-20 (around 0.2-0.3 for most datasets).We use a concept vocabulary chosen from a subset of LAION tokens as described in Section 4.1. Both image embeddings and dictionary concepts are centered and normalized as mentioned in Section 4.1, with the image mean used for centering computed over the MSCOCO train set and the concept mean computed over our chosen vocabulary.

### 5.2 Sparsity-Performance Tradeoffs

We assess the performance of SpLiCE decompositions by evaluating the reconstruction error in terms of cosine similarity between SpLiCE representations and CLIP embeddings, the zero-shot performance of SpLiCE decompositions, and the retrieval performance of SpLiCE embeddings. We compare the performance of decompositions generated from our semantic concept vocabulary to decompositions over random vocabulary and learned dictionary vocabulary baselines. All vocabularies are of size 15,000 concepts. The random vocabulary is sampled from a 512-dimensional normalized Gaussian distribution. The learned vocabularies are generated by using the Fast Iterative Shrinkage-Thresholding Algorithm (FISTA) [62] to learn optimal dictionaries given our sparse recovery problem (optimizing Equation (1) for both $\mathbf{C}$ and $w$). Note that we learn separate dictionaries $\mathbf{C}_{\text{img}}$ and $\mathbf{C}_{\text{text}}$ to reconstruct MSCOCO image and text embeddings respectively. In Figure 3, we plot the cosine reconstruction and zero-shot accuracy of image decompositions with the various dictionaries. We evaluate probing performance (Tables 3, 4) and text-to-image and image-to-text retrieval in the Appendix (Figure B.3).

These results overall show SpLiCE efficiently navigates the interpretability-accuracy Pareto frontier and retains much of the performance of black-box CLIP representations with the semantic, human-interpretable LAION dictionary, significantly outperforming other dictionaries on semantic tasks such as zero-shot classification, probing, and retrieval. At the same time, we find that our semantic LAION dictionary does not result in accurate cosine similarity reconstructions of the original CLIP, often being on par with using random dictionaries. We believe this is because CLIP encodes both semantics of the underlying image and non-semantic "noise", which violates Assumption #2 in Section 3. Given that our SpLiCE decompositions only aim to encode semantics, they are unable to encode non-semantic aspects in the underlying representation, thus causing poor alignment in the cosine similarity sense, while simultaneously exhibiting excellent alignment on semantic tasks such as zero-shot accuracy. For ImageNet, we find that many classes are animal species that cannot easily be described by 1-2 words (e.g. 'red-breasted merganser', 'American Staffordshire terrier'). Adding these class labels to our concept dictionary increases performance significantly, as shown by the dotted yellow line in Figure 3.

---

[3]For more discussion on the relationship between Eq. (1) and Eq. (2), see Appendix, Sec. A.2

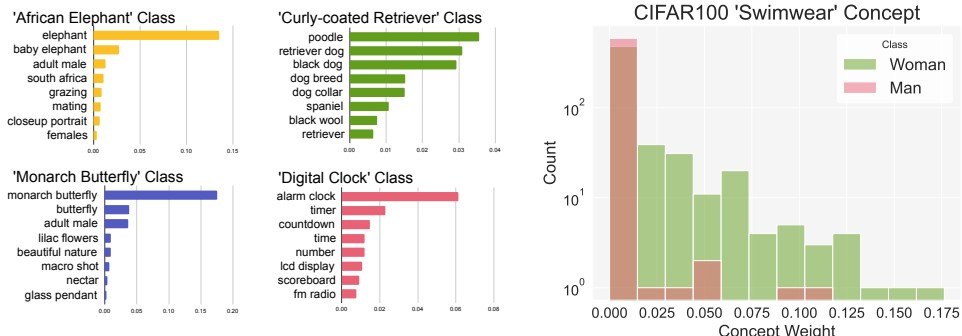

Figure 4: **Left:** SpLiCE decompositions of ImageNet 'African Elephant', 'Curly-coated Retriever', 'Monarch Butterfly', 'Digital Clock' classes. **Right:** Distribution of "Swimwear" concept in 'Woman' and 'Man' classes of CIFAR100.

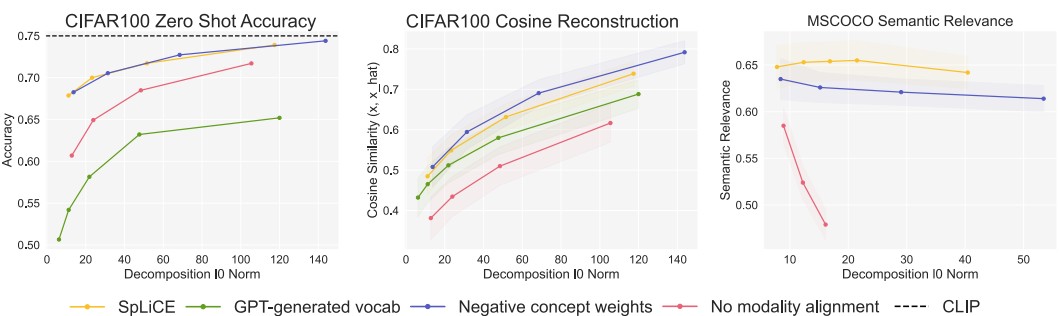

Figure 5: Ablation study evaluating the efficacy of SpLiCE design choices across three metrics: Zero-shot accuracy, cosine reconstruction, and semantic relevance of recovered tags. We find that all of our design choices, namely non-negativity, modality alignment, and usage of large task-agnostic dictionary are essential to performance. See §5.3 for discussion.

## 5.3 Ablation Studies

We perform ablation studies to evaluate the effectiveness of the design decisions of SpLiCE, including the choice of vocabulary, the nonnegativity of the decompositions, and the modality alignment by ablating each choice and observing the effect on three metrics: zero-shot accuracy on CIFAR100, cosine similarity between the reconstructed and original embeddings of CIFAR100, and semantic relevance on MSCOCO. The first two metrics are the same as those presented in Figure 3. We compute semantic relevance by tokenizing and filtering stop-words from the MSCOCO human-generated captions and embedding each token with CLIP. Then, we take all non-zero concepts output by SpLiCE and compute the Hausdorff distance between the sets of SpLiCE concepts and caption token embeddings. This essentially measures how aligned decompositions are with human captions. We observe that replacing our dictionary with the LLM-generated concept dictionary used by [22, 24, 23] significantly worsens the decomposition in terms of zero shot accuracy and cosine reconstruction. While allowing for negative concept weights improves cosine reconstruction marginally, it decreases the semantic relevance of the decompositions, as negative concepts frequently correspond to concepts not present in images, and as such, are unlikely to be represented by human captions. Finally, we see that modality alignment is necessary across all three metrics. Overall, these ablation studies show that each aspect of SpLiCE is necessary for creating human-interpretable, semantically relevant and highly performant decompositions.

## 5.4 Qualitative Assessment of Decompositions

**Concept Decompositions for Images.** We visualize SpLiCE decompositions to qualitatively assess the semantic content of the images they represent. In Figure 2 we provide six sample decompositions from MSCOCO with their corresponding captions. We display the top seven concepts for each

image and find that they generally well describe the semantics of the images. We also find that these qualitative examples yield interesting and unexpected insights into both CLIP and the data. In the top left image, we see that the decomposition includes the text present on the sign in the image, revealing that CLIP prioritizes text in images over objects. For the bottom left image, the decomposition correctly includes the concept "`macro shot`", revealing that CLIP encodes information regarding geometric perspective. The bottom right decomposition similarly features the concept "`blackandwhite bw`", indicating that CLIP encodes not only the objects present in images but also information about the lighting and color. Overall, these results suggest that SpLiCE may also be used as a zero-shot image tagging method to understand images.

**Concept Histograms for Datasets.** Beyond concept-based explanations of individual images, we propose that SpLiCE can be used to better understand and summarize collections of images, such as entire datasets. To compute concept decompositions of sets of images, we decompose each individual image and aggregate the results, which we use to generate concept histograms of the dataset. We visualize four concept histograms for the ImageNet classes '`African Elephant`', '`Curly-coated Retriever`', '`Monarch Butterfly`', and '`Digital Clock`', in Figure 4. These decompositions provide information about the distribution of the data and how CLIP represents it. For example, digital clocks are differentiated from analog clocks through the concepts "`lcd display`" and "`countdown`". Monarch butterflies are highly correlated with the concept "`lilac flowers`" in ImageNet, which we validated through manual inspection (nearly half of the monarch butterfly images in the validation set feature purple flowers). Interestingly, '`Curly-coated retrievers`' are represented as combinations of "`poodle`", "`retriever dog`", and "`black dog`", which perfectly describe the main characteristics of them: black retrievers with poodle-textured fur.

# 6    Case Studies and Applications of SpLiCE

In this section, we present two example case studies using SpLiCE: (1) spurious correlation and bias detection in datasets and (2) debiasing classification models. We present additional case studies for (1) and (2), as well as (3) monitoring distribution shift in Appendix B.6, B.7, B.8 B.9. We also present results from a user study to evaluate the human interpretability of SpLiCE in Appendix B.1, where we find that users prefer explanations generated by SpLiCE over existing Concept Bottleneck Model-based methods.

**Discovering Spurious Correlations in CIFAR100.**    Existing methods to detect spurious correlations in datasets generally require subgroup and attribute labels or rely on manual human inspection of images (see [63] for an overview), making it hard to scale to large datasets. SpLiCE, on the other hand, allows for fast automatic detection of such biases, without any labels, training, or even a task. To illustrate this, we study two classes of CIFAR100: 'man' and 'woman', in Figure 4. Upon decomposing these classes, we found that {"`bra`","`swimwear`" } were two of the top ten most common concepts in the 'woman' class. On the other hand, the only clothing-related concepts that appear in the top 50 most activated concepts for 'man' are {"`uniform`", "`tuxedo`", "`apparel`"}. We visualize a histogram of the concept weights on swimwear- and undergarment-related concepts {"`swimwear`", "`bra`", "`trunks`", "`underwear`"} across both the train and test sets, and find that these concepts are much more likely to be activated for women than men. Manual inspection of CIFAR100 verifies the trend highlighted by SpLiCE, where *at least 70 of the 600 images in the 'woman' class feature women in bikinis, underclothes, or even partially undressed*, revealing stereotype bias in this popular dataset. We provide a similar study of the concept "`desert`" with respect to the 'camel' and 'kangaroo' classes in CIFAR100 in Appendix B.6.

**Model Editing on CelebA Attribute Classifiers.**    Concept-based representations unlock a key application: being able to intervene on and edit models. This edit can be performed in two equivalent ways: either on the concept representations themselves, where we can zero out a concept or on linear probes built upon the decompositions, where we can edit the weight matrix between concepts and class labels (similar to concept bottleneck models [18]). Here, we evaluate the efficacy of SpLiCE for these forms of model editing. Specifically, we consider two tasks on CelebA, classifying gender and whether the subject is wearing glasses. To test representation editing, we remove the concept of "eyewear" or "glasses" from CelebA image representations by zeroing out any weight placed on these concepts in our SpLiCE decompositions and evaluate classifier performance. We report the performance of zero-shot classification and linear probes over our SpLiCE representation in Table

Table 2: Evaluation of intervention on the concept 'Glasses' for the CelebA dataset. SpLiCE allows for surgical removal of information related to whether or not someone is wearing glasses, without impacting other features such as gender. (ZS = Zero Shot Accuracy)

|  | GENDER | GLASSES |
|---|---|---|
| ZS CLIP | 0.98 | 0.91 |
| ZS SPLiCE | 0.97 | 0.88 |
| ZS INTERVENTION SPLiCE | 0.96 | **0.69** |
| LINEAR PROBE | 0.89 | 0.88 |
| INTERVENTION PROBE | 0.85 | **0.59** |

2. In both cases, we find that we can surgically target and remove information pertaining to glasses and reduce classifier performance while preserving information relevant to gender classification. We perform a similar experiment on the Waterbirds dataset [64] to remove spurious background signals in B.7.

# 7 Discussion

In this work, we show that the information contained in CLIP embeddings can be approximated by a sparse, linear combination of simple semantic concepts, allowing us to interpret representations via sparse recovery. We propose SpLiCE, a method to transform the dense, uninterpretable embeddings of CLIP into human-interpretable sparse concept decompositions.

We empirically demonstrate that SpLiCE allows for an adjustable tradeoff on the interpretability-accuracy Pareto frontier, enabling users to decide the loss in performance they are willing to incur for interpretability. Furthermore, we find that the improved interpretability of SpLiCE allows for users to diagnose and fix model mistakes, ideally increasing the effectiveness and performance of the overall system using a VLM. We then provide concrete use cases for SpLiCE: spurious correlation detection and model intervention and editing, showcasing the benefits of using interpretable embeddings with known semantic content. We highlight that SpLiCE embeddings can serve as post-hoc interpretations of CLIP embeddings and can even replace them to ensure full transparency.

**Broader Impact.** Similar to many works in the field of interpretability, our work provides greater understanding of the behavior of models, including but not limited to the broader implicit biases they perpetuate as well as mistakes made on individual samples. We believe this is particularly salient for CLIP, which is used in a variety of applications that are widely used in practice at this moment. We hope that insights gained from such interpretability allow users to make more informed decisions regarding how they interact with and use CLIP, regardless of their familiarity with machine learning or domain expertise in the task they are using CLIP for. We also highlight that SpLiCE can be used as a visualization-like tool for exploring and summarizing datasets at scale, allowing for easier auditing of spurious correlations and biases in both datasets and models.

**Limitations.** In this work, we use a large, overcomplete dictionary of one- and two-word concepts, however future work may wish to expand this dictionary or learn a dictionary over tokens (in discrete language space), to capture concepts with more than two words. This may also reduce the size of the dictionary and improve computation time. We note that this dictionary was constructed by looking at token frequency in the LAION-5B dataset, which has its own biases and may not correctly capture all the salient concepts that CLIP encodes. Despite this, we find that SpLiCE performs well on a variety of tasks while outperforming state-of-the-art concept dictionaries (Fig. 5, Appendix Fig. 13) and thus we believe LAION is a good dataset to generate a concept vocabulary from. We also note that this vocabulary can be easily modified by practitioners to consider additional concepts as needed for specific use cases. Finally, SpLiCE also uses an $\ell_1$ penalty as the relaxation for $\ell_0$ regularization, but future work may consider alternative relaxations or even binary concept weights.

## Acknowledgements and Disclosure of Funding

This work is supported in part by the NSF awards IIS-2008461, IIS-2040989, IIS-2238714, FAI-2040880, and research awards from Google, JP Morgan, Amazon, Adobe, Harvard Data Science Initiative, and the Digital, Data, and Design ($D^3$) Institute at Harvard. AO is supported by the National Science Foundation Graduate Research Fellowship under Grant No. DGE-2140743, and UB is funded by the Kempner Institute Graduate Research Fellowship. The views expressed here are those of the authors and do not reflect the official policy or position of the funding agencies.

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

# Appendix

## Summary of Appendix Results

## A    Further Details on the Method

### A.1    When do Sparse Decompositions Exist?

**Proposition 1.** *Given Assumptions 1-5, CLIP image embeddings $f$ can be written as a sparse linear combination of text embeddings, i.e,*

$$f(\mathbf{x}^{img}) = \mathbf{C}^{txt}\mathbf{w}; \ \ s.t. \ \|\mathbf{w}\|_0 \leq \alpha$$

*where $\mathbf{w} \in \mathbb{R}_+^k$, and $\mathbf{C}^{txt} \in \mathbb{R}^{d \times k}$, which is the text concept dictionary defined previously.*

*Proof.* Any vector $\omega$ can be written as $\omega = \sum_{i=1}^k \omega_i \mathbf{e}_i$, where $\omega_i \in \mathbb{R}_+$, and $\mathbf{e}_i \in \mathbb{R}^k$ is a one-hot vector with one at the $i^{\text{th}}$ co-ordinate. Thus we have

$$f(\mathbf{x}^{img}) = f \circ h^{img}(\omega, \epsilon) = f \circ h^{img}(\omega) \quad \text{(Assumption 2)}$$

$$= f \circ h^{img}\left(\sum_{i=1}^k \omega_i \mathbf{e}_i\right) = \sum_{i=1}^k \omega_i \underbrace{f \circ h^{img}(\mathbf{e}_i)}_{\mathbf{c}_i^{img}} \quad \text{(Assumption 3)}$$

Here we define $\mathbf{c}_i^{img} = f \circ h^{img}(\mathbf{e}_i)$ as the 'image' concept basis vector; analogous to the text concept basis vector $\mathbf{c}_i^{txt} = g \circ h^{txt}(\mathbf{e}_i)$ already defined. Thus Assumption 2 implies the existence of a sparse decomposition of $f$ in terms of 'image' concept vectors $\mathbf{c}_i^{img}$. Additionally, Assumption 1 ensures that this decomposition is sparse, as $\omega$ is sparse. So far, we have $f(\mathbf{x}^{img}) = \mathbf{C}^{img}\omega \ \ s.t. \ \|\omega\|_0 \leq \alpha$.

From Assumption 4, the image concept vectors and text concept vectors are equal to each other, i.e, $\mathbf{c}_i^{img} = f \circ h^{img}(\mathbf{e}_i) = g \circ h^{txt}(\mathbf{e}_i) = \mathbf{c}_i^{txt}$. Finally, from Assumption 5, we have that the text concept vectors $\mathbf{c}_i^{txt}$ are given simply by word embeddings $g$ of individual words.

Stringing these arguments together, we have that image representations $f(\mathbf{x}^{\text{img}})$ can be written as a sparse linear combination of vectors obtain from CLIP word embeddings $\mathbf{c}_i^{\text{txt}}$. We finally set $\mathbf{w} = \omega$, thus proving the assertion.

$\square$

## A.2 Relationship between cosine similarity and MSE optimization.

Recall our $\ell_1$ relaxed cosine similarity optimization problem from Eqn. (1),

$$\min_{\mathbf{w} \in \mathbb{R}_+^c} \|\mathbf{w}\|_0 \ \text{ s.t. } \ \langle \mathbf{z}, \frac{\mathbf{Cw}}{\|\mathbf{Cw}\|_2} \rangle \geq 1 - \epsilon. \tag{3}$$

First we relax the $\ell_0$ constraint to an $\ell_1$ penalty.

$$\max_{\mathbf{w} \in \mathbb{R}_+^c} \langle \mathbf{z}, \frac{\mathbf{Cw}}{\|\mathbf{Cw}\|_2} \rangle - \lambda \|\mathbf{w}\|_1. \tag{4}$$

By observing that $\|x - y\|_2^2 = \langle x - y, x - y \rangle = \langle x, x \rangle + \langle y, y \rangle - 2\langle x, y \rangle$ and that $\mathbf{z}, \frac{\mathbf{Cw}}{\|\mathbf{Cw}\|_2}$ are unit-norm, maximizing the above inner product is equivalent to minimizing the euclidean norm,

$$\min_{\mathbf{w} \in \mathbb{R}_+^c} \|\frac{\mathbf{Cw}}{\|\mathbf{Cw}\|_2} - \mathbf{z}\|_2^2 + 2\lambda \|\mathbf{w}\|_1. \tag{5}$$

This is a non-convex problem, but we can relax this problem to achieve better reconstruction in terms of euclidan distance as shown in Eqn. (2),

$$\min_{\mathbf{w} \in \mathbb{R}_+^c} \|\mathbf{Cw} - \mathbf{z}\|_2^2 + 2\lambda \|\mathbf{w}\|_1. \tag{6}$$

This problem will optimize euclidean distance between $\mathbf{Cw}$ and $\mathbf{z}$. Consider two vectors $x, y$ on the unit sphere such that $\langle \frac{x}{\|x\|}, \frac{y}{\|y\|} \rangle > 0$. While any vector $\alpha y, \alpha > 0$ will have the same cosine similarity score, the optimal vector in terms of euclidean distance to $x$ is the vector $\alpha y$ such that $\alpha = \text{proj}_y(x)$, or in other words the projection of $x$ onto $y$. Thus, solving for euclidean distance to approximate $x$ will find $\alpha y$ which we must then normalize to find the unit-norm solution $y$. This explains the normalizing process described in Section 4.1.

Additionally, we can view Eqn. (6) as applying shrinkage to $\mathbf{Cw}$. Reconverting from euclidean norm to inner product, Eqn. (6) becomes

$$\max_{\mathbf{w} \in \mathbb{R}_+^c} \langle \mathbf{Cw}, \mathbf{z} \rangle - \frac{1}{2}\langle \mathbf{Cw}, \mathbf{Cw} \rangle - \lambda \|\mathbf{w}\|_1 = \langle \mathbf{Cw}, \mathbf{z} \rangle - \frac{1}{2}\|\mathbf{Cw}\|_2^2 - \lambda \|\mathbf{w}\|_1. \tag{7}$$

In conclusion, our optimization problem maximizes the inner product while imposing a shinkage penalty and sparsity penalty. Empirically, our reconstructions $\mathbf{Cw}$ are low-norm, so we normalize after solving to recover the unit-norm reconstruction.

## A.3 ADMM for batched on-device LASSO optimization.

As each decomposition requires solving a LASSO optimization problem, we implement the Alternating Direction Method of Multipliers (ADMM) algorithm in Pytorch over batches with GPU support for efficient decomposition of large scale datasets over large numbers of concepts [65]. In practice, ADMM achieves primal and dual tolerances of $1\text{e} - 4$ in fewer than $1000$ iterations on a batch size of $1024$. We present an empirical comparison beterrn LASSO and ADMM in 6, where we find both methods to be approximately equivalent.

Next we derive the iterates for our ADMM algorithm. Recall our optimization problem,

$$\min_{\mathbf{w} \in \mathbb{R}_+^c} \|\mathbf{Cw} - \mathbf{z}\|_2^2 + 2\lambda \|\mathbf{w}\|_1. \tag{8}$$

ADMM breaks down convex optimization problems into multiple sub-problems while penalizing the difference in solutions. We break Eqn. (8) into two subproblems, one solving the euclidean distance

objective and one solving the $\ell_1$ and nonnegativity constraint. We let $w$ denote the former solution, $z$ the latter, and $u$ tracks the difference between the two. Our ADMM iterates $(w^k, z^k, u^k)$ are

$$w^{k+1} = \arg\min_w (f(w) + \frac{\rho}{2}||w^k - z^k + u^k||_2^2), \tag{9}$$

$$z^{k+1} = (S_{\lambda/\rho}(w^{k+1} + u^k))_+, \tag{10}$$

$$u^{k+1} = u^k + w^{k+1} - z^{k+1}, \tag{11}$$

where $S_\kappa$ is a soft-thresholding function used to satisfy the LASSO constraints,

$$S_\kappa(a) := \begin{cases} a - \kappa, & a > \kappa \\ 0, & |a| \leq \kappa \\ a + \kappa, & a < -\kappa \end{cases} \tag{12}$$

As our optimization function $f(w)$ is quadratic, we can analytically compute $w^{k+1}$ as

$$w^{k+1} = (2\mathbf{C}^T\mathbf{C} + \rho)^{-1}(\rho v + 2\mathbf{C}w), \tag{13}$$

where $v = z^k - u^k$. In our experiments we set $\rho = 5$, and stop when tolerances $\epsilon_{\text{prim}} = ||x^{k+1} - z^{k+1}||_2$, $\epsilon_{\text{dual}} = ||\rho(z^{k+1} - z^k)||_2$ are less than $1e - 4$. Over a batch, we iterate until every solver in the batch has reached the above tolerances.

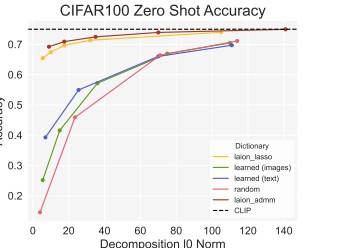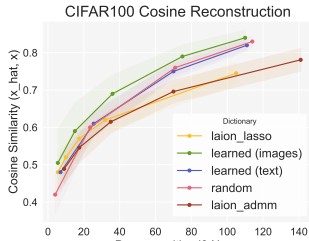

Figure 6: Comparison of ADMM (maroon) and LASSO (yellow) for solving the SpLiCE objective on zero shot accuracy (left) and cosine reconstruction (right) on CIFAR100. Both methods are approximately equal.

## A.4 Effect of Modality Alignment

We take MSCOCO images and captions, embed them with CLIP, and compare the cosine similarity between modalities and inter-modality. Before mean-centering and renormalizing, the similarity within modalities is high, with an average of around 0.3. This indicates that the image and text embeddings do not span the entire unit-sphere but rather lie on two cones. However, the similarity across modalities has an average concentrating around zero, indicating that these two cones are non-overlapping. However, after mean-centering and normalizing, we observe that the average cosine similarity for images, text, and between images and text becomes zero and the modalities are aligned.

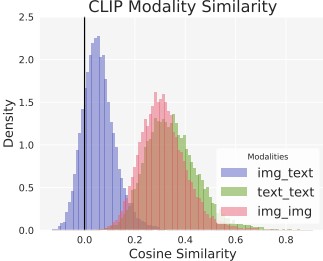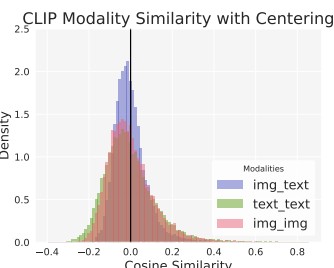

Figure 7: Average cosine similarity across pairs of image-text, image-image, and text-text data from MSCOCO. After aligning modalities, the distribution of similarities is centered around zero.

## A.5 Experimental Details

All experiments are able to be performed on a single A100 GPU to run fast inference with CLIP. After embedding the concept dictionary, all computation can be performed on a CPU. Code is made available at `https://github.com/AI4LIFE-GROUP/SpLiCE`.

# B Additional Results

## B.1 User Study for Human Interpretability

We present results from a user study in 8 to assess the human interpretability of SpLiCE. We base our study off of that performed by [23] to evaluate Label-Free Concept Bottleneck Models (LF-CBMs). We benchmark our method against LF-CBMs and IP-OMP [22]. We provided users with twenty randomly chosen, correctly predicted images from ImageNet and explanations from two different methods comprising the top six most important concepts for every image. We then asked users to evaluate and compare the different concept-based explanations for (1) their relevance to the provided image inputs, (2) their relevance to model predictions, and (3) their informativeness on Likert scales from 1 to 5. We found that users significantly preferred explanations generated by SpLiCE to the two baselines for relevance to the images and informativeness, with significance determined via a one-sample two-sided t-test and a threshold of p=0.01. We also highlight that our method is able to produce similar/better concept decompositions, in terms of human interpretability, than the baselines without needing to train a classification probe or use class labels for concept mining, both of which are computationally expensive. This user study was ruled exempt by our institution's IRB, as no risks were posed to the users. Participants were able to opt out at any time, and no questions were asked regarding the participants themselves.

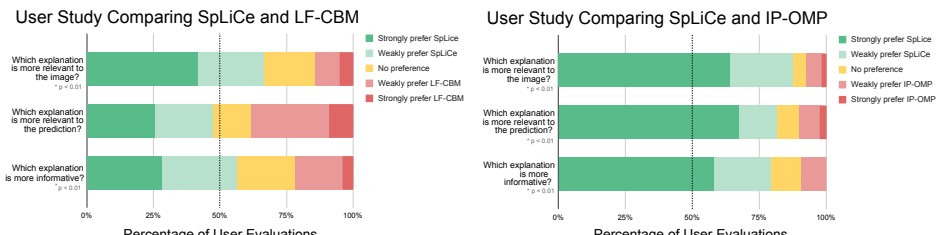

Figure 8: Results of a user study evaluating SpLiCE, LF-CBM, and IP-OMP in the style of the user study from LF-CBM. Overall, we find that explanations generated by SpLiCE are deemed more relevant to the image, relevant to the prediction, and more informative than prior methods.

## B.2 Performance of SpLiCE on Probing Tasks

We evaluate the performance of the decompositions on probes trained on both regular CLIP embeddings as well as decomposed CLIP embeddings for CIFAR100 in 3 and MIT States in 4. We consider two scenarios: a probe trained on CLIP embeddings and tested on SpLiCE embeddings of various sparsities (shown in row `CLIP Probe`), and a probe both trained and evaluated on SpLiCE embeddings (shown in row `SpLiCE Probe`). We report mean over three runs, with standard deviations for each experiment being less than 0.005. We find that SpLiCE representations closely match the performance of dense CLIP embeddings, with a slight drop in performance when probes are trained directly on SpLiCE embeddings rather than trained on CLIP embeddings and evaluated on SpLiCE embeddings for CIFAR100.

Table 3: Evaluation of Probing Performance on CIFAR100

|  | $l_0 = 3$ | $l_0 = 6$ | $l_0 = 23$ | $l_0 = 117$ | CLIP |
|---|---|---|---|---|---|
| SPLICE PROBE | 0.95 | 0.95 | 0.95 | 0.95 | – |
| CLIP PROBE | 0.96 | 0.96 | 0.97 | 0.97 | 0.97 |

Table 4: Evaluation of Probing Performance on MIT States

|  | $l_0 = 4$ | $l_0 = 7$ | $l_0 = 27$ | CLIP |
|---|---|---|---|---|
| SPLICE PROBE | 0.883 | 0.883 | 0.882 | – |
| CLIP PROBE | 0.883 | 0.883 | 0.884 | 0.883 |

## B.3  SpLiCE Performance on Retrieval Tasks

We test the performance of SpLiCE embeddings on text-to-image and image-to-text retrieval tasks. We evaluate retrieval over various 1024 sample subsets of MSCOCO, and assess recall performance for the top-k closest embeddings of the opposite modality for $k = \{1, 5, 10\}$. We find that our semantic concept dictionaries outperform all baselines when decomposition sparsity is high, but that dictionaries learned over images perform slightly better for text to image retrieval when decompositions have greater than 30 nonzero concepts.

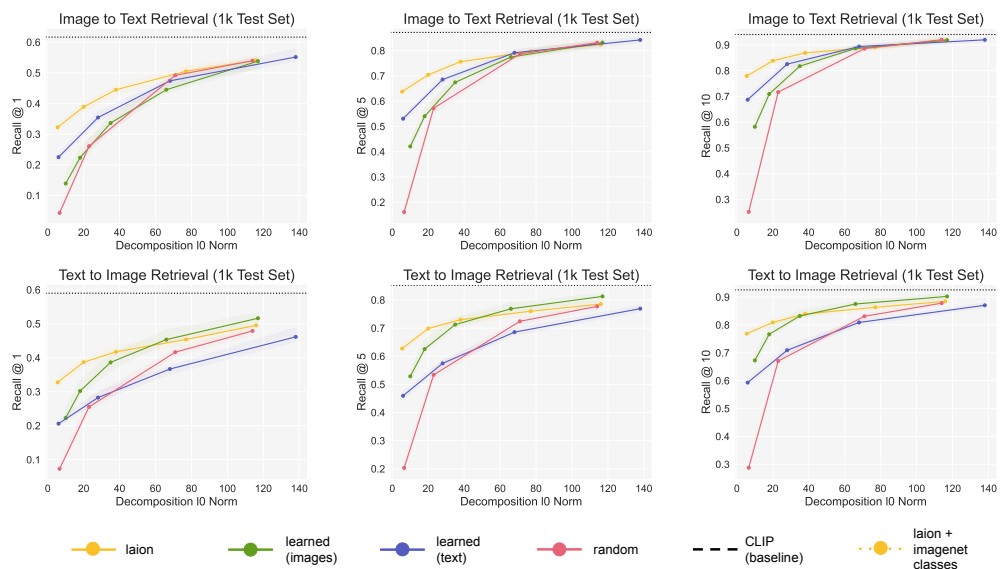

Figure 9: Top-1 , 5, 10 performance of SpLiCE representations on image-to-text (top) and text-to-image (bottom) retrieval on MSCOCO.

## B.4  Additional Zero-Shot Results

We present additional results comparing SpLiCE reconstructed vectors and CLIP embeddings on the Caltech101 [66], SUN397 [67], STL10 [68], and VOC2007 [69] datasets in 5. We use SpLiCE decompositions with sparsities of 20-35, and we find that they are comparable to the unaltered CLIP embeddings.

Table 5: Additional zero-shot accuracy on baselines from the CLIP paper, for decompositions of sparsity 20-35. Note that at human-interpretable levels of sparsity, we see a minor drop in performance.

|  | CALTECH101 | SUN397 | STL10 | VOC 2007 |
|---|---|---|---|---|
| CLIP REPORTED | 0.88 | 0.63 | 0.97 | 0.83 |
| CLIP IMPLEMENTED | 0.90 | 0.67 | 0.96 | 0.92 |
| SPLICE | 0.86 | 0.66 | 0.96 | 0.83 |

We further explore the performance of SpLiCE decompositions in the limit as they approach the sparsity of the baseline CLIP embeddings (512). We find that SpLiCE completely recovers CLIP zero-shot accuracy at this limit, as shown in 6.

Table 6: Zero shot performance at sparsity 512. Note that SpLiCE completely recovers baseline CLIP zero shot accuracy.

|  | CIFAR100 | MITSTATES | IMAGENET |
|---|---|---|---|
| CLIP BASELINE | 0.750 | 0.469 | 0.552 |
| SPLiCE (512) | 0.768 | 0.474 | 0.552 |

## B.5    Additional ImageNet Concept Histograms

We present concept histograms for the top seven concepts of five more ImageNet classes: {'Face Powder', 'Feather Boa', 'Jack-O'-Lantern', 'Kimono', 'Dalmation'}, similar to Figure 10. These decompositions give insights both into the distribution of each class as well as some biases of CLIP. For example, for the class 'Face Powder', the concept "benefit" is the fifth most common concept, and it is indeed a common cosmetic brand name in the images. For the 'Dalmation' class, we see that the decompositions consists of concepts relating to dogs and black and white spots, which together make up the high-level concept of a dalmation. Finally, for the class 'Kimono', the concept "doll" is the seventh most common, although all of the images in the 'Kimono' class were of real humans, not of dolls. This highlights an implicit bias in CLIP's representations or in the descriptions of people wearing kimonos in CLIP's training set.

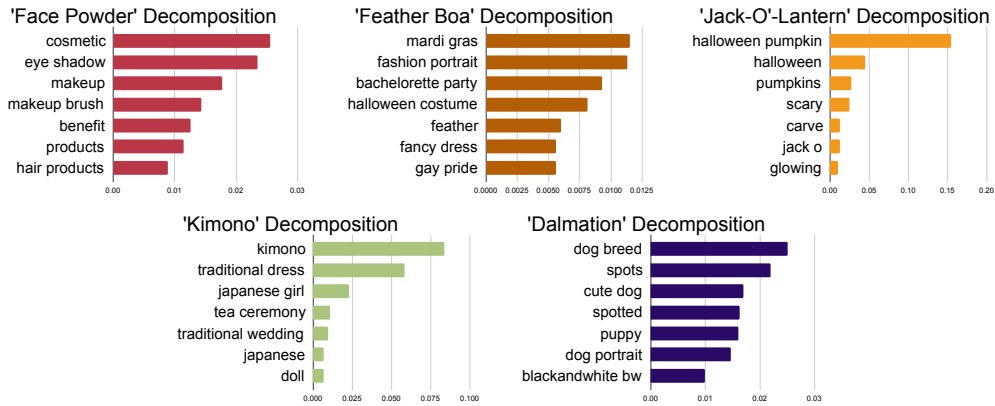

Figure 10: Example concept histograms of various ImageNet classes. The top seven concepts for each class are visualized along with their relative weighting, with the average $\ell_0$ norm of individual sample decompositions also being 7.

## B.6    Additional Case Study: Detecting Spurious Correlations

We present an additional case study for detecting spurious correlations in CIFAR100. In particular, we look at the prevalence of the spurious concept "desert" in the classes 'camel' and 'kangaroo' in Figure 11. We observe that camels are more frequently pictured in the desert, creating a spurious signal that may be leveraged by downstream classifiers. This figure provides an additional example of how we can understand biases and trends in data with SpLiCE decompositions.

## B.7    Additional Case Study: Spurious Correlation Intervention

We further test the ability of SpLiCE to enable intervention on intermediate representations and linear classifiers by attempting to remove information pertaining to spurious signals. In particular, we consider the Waterbirds dataset [64], which spuriously correlates landbirds with land backgrounds, resulting in trained classifiers performing poorly on waterbirds on land. We thus remove information about whether or not birds are on land backgrounds by ablating concept weights on "bamboo",

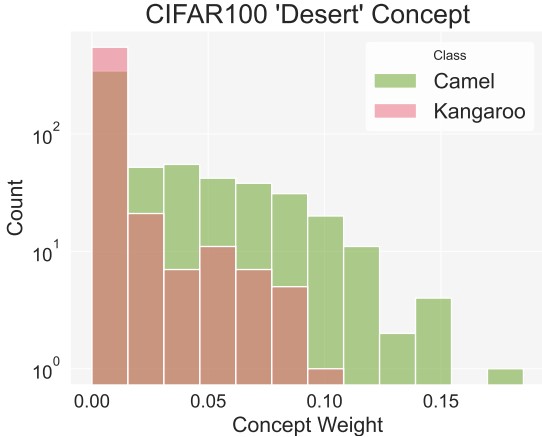

Figure 11: Distribution of "Desert" concept in 'Camel' and 'Kangaroo' classes of CIFAR100.

Table 7: Evaluation of intervention on spurious correlations for Waterbirds dataset. Removing information about land backgrounds improves worst-case subgroup performance.

|  | LANDBIRDS ON LAND | WATERBIRDS ON LAND |
| --- | --- | --- |
| LINEAR PROBE | 0.98 | 0.48 |
| INTERVENTION PROBE | 0.97 | **0.60** |

"forest", "hiking", and "rainforest" as well as any bigrams containing the word "forest," as shown in Table 7. This significantly improves worst-case subgroup performance for waterbirds on land from 0.48 to 0.60.

For both this experiment and the intervention on CelebA described in the main paper, we train linear probes using the LogisticRegressionClassifier module in scikit-learn using an $\ell_1$ penalty.

### B.8 Additional Case Study: Distribution Shift Monitoring

We present a final case study using SpLiCE to monitor distribution shift. This can help identify differences between training and inference distributions or evaluate how a continually sampled dataset changes over time. In this experiment we consider the Stanford Cars dataset [70], which contains photos of cars from 1991 to 2012, including their make and year labels. By decomposing photos of cars from each year, we can view how the distribution changed yearly. We visualize the weights of the concepts "convertible" and "yellow" from our decompositions, as well as the actual percentage of cars from each year that were convertibles or yellow in Figure 12. Note the right-hand y-axis, corresponding to the weight of the given concept $c_i$ over the sum of the weights of all concepts $\sum_i c_i$, does not have a meaningful unit of measure or scale. We find that the trends in the groundtruth concept prevalence generally closely match that of the predicted/decomposed concepts, allowing us to visualize which years convertibles or yellow cars were popular or out-of-distribution with respect to other years. Most notably, we see that SpLiCE picks up on the out-of-distribution rise in popularity of brightly colored sports cars in the early 2000s.

### B.9 Additional Case Study: Distribution Shift Monitoring

To further verify that SpLiCE allows for identification and tracking of distribution shift, we study the Waterbirds dataset, which is known to have differently balanced train, valodation, and test splits. To identify distribution shifts, we can simply look at the norm of the difference between the class decompositions of the two classes for each splot, as shown in 8. We find that the validation and test splits are much more similar than the training and validation splits or the training and test splits, which can be verified by the construction process of the Waterbirds dataset.

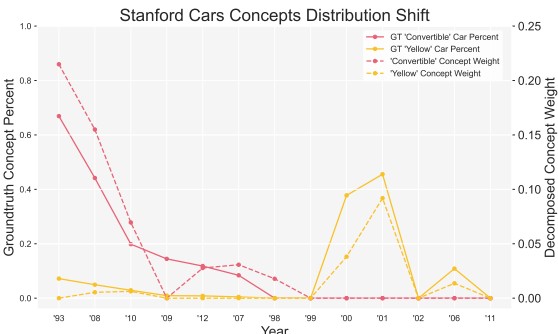

Figure 12: Visualization of the presence of convertibles (pink lines) and yellow cars (yellow lines) in Stanford Cars over time. SpLiCE concept weights (dotted) closely track the groundtruth concept prevalence (solid) for both concepts.

Table 8: Study of the differences in distributions between train, validation, and test splits of Waterbirds. The validation and test splits are much more similar to each other than they are to the train split.

|  | TRAIN, VAL | TRAIN, TEST | VAL, TEST |
|---|---|---|---|
| CLASS LANDBIRD | 0.0182 | 0.0182 | **0.005** |
| CLASS WATERBIRD | 0.0229 | .0188 | **0.009** |

We also find that the most weighted concept in the 'landbird' class of the train split is "bamboo" but the corresponding weight for "bamboo" in the 'waterbird' class is much lower. The "bamboo" concept weight for both classes and all splits is shown below, where we see that the validation and test splits are very similar and mostly evenly balanced, whereas the train split is highly unbalanced.

### B.10 Checking the Interpretability of Negative Concepts

We take a set of 71 concept-antonym pairs from the MIT States dataset and embed the terms in CLIP. With and without concept centering, we observe that these concept-antonym pairs have an average cosine similarity well above -1, indicating that CLIP does not place antonyms in opposite directions, as shown in 10. Next, we take our concept dictionary and prepend "not" to all of the words and compare the average cosine similarity between concept and not-concept pairs. Similarly, we observe that with and without centering, concept and not-concept pairs are highly similar. Note that the average similarity for true pairs of images and text in MSCOCO is less than the similarity between concepts and not-concepts with and without centering.

### B.11 Understanding the Image Mean for Modality Alignment

In order to empirically check that the mean centering of images does not result in a loss of information, we decompose the img mean, $\mu_{img}$, that we used for all experiments. If we decompose it with uncentered concepts, the following concepts are highlighted: {"closeup", "flickr", "posed"}. The decomposition with centered concepts results in the following concepts: {"flickr", "posed", "pics", "angle view", "last post"}. These concepts all seem to be generally related to images, with minimal other semantic information, suggesting that centering does not remove any discriminative semantic content of embeddings, but simply removes information about the modality.

Table 9: Study of the prevalence of the concept "bamboo" in the different classes and splits of Waterbirds.

|  | TRAIN | VAL | TEST |
|---|---|---|---|
| CLASS LANDBIRD | **0.0196** | 0.010 | 0.010 |
| CLASS WATERBIRD | **0.0007** | 0.008 | 0.008 |

Table 10: Evaluation of the similarity of antonyms and negative concepts in CLIP.

| | PAIRWISE COSINE SIMILARITY (WITHOUT CONCEPT CENTERING) | PAIRWISE COSINE SIMILARITY (WITH CONCEPT CENTERING) |
|---|---|---|
| CONCEPT AND ANTONYM | $0.7176 \pm 0.1109$ | $0.1366 \pm 0.2197$ |
| CONCEPT AND "NOT" CONCEPT | $0.8661 \pm 0.0498$ | $0.6130 \pm 0.0498$ |

## B.12 Choice of Concept Vocabulary

We perform a simple ablation study to assess the sensitivity of our method to choices in concept vocabulary. We collect a second vocabulary in the same exact manner as the LAION vocabulary from the MSCOCO caption dataset. We consider both the top 10k and top 5k most common words for both, and repeat the zero-shot accuracy and reconstruction cosine similarity experiments from Section 5.2 on CIFAR100. We see that the MSCOCO10k and LAION10k vocabularies perform almost exactly the same for both metrics. The smaller vocabularies perform the same for cosine reconstruction but underperform the 10k vocabularies for zero-shot classification tasks.

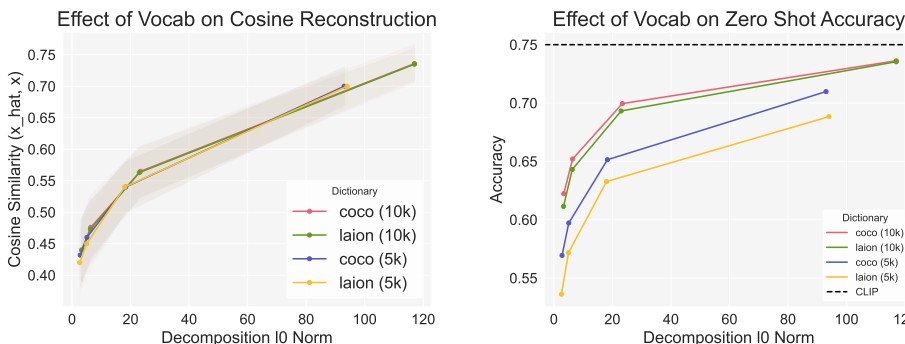

Figure 13: Change in SpLiCE performance when considering another semantic concept dictionary derived from MSCOCO as well as a smaller concept vocabulary.

## B.13 Concept Type Distribution

In order to better understand any biases produced by the decomposition process or that CLIP itself has, we visualize the types of concepts most commonly activated across multiple datasets, labelling them by part of speech in Figure 14. We see that nouns are by far the most common concepts across datasets, indicating that both CLIP and the decompositions are highly object centric. Note that the low weight on verbs and adjective is due to far fewer concepts of those types being activated (low $l_0$ norm) as well as the weight upon those concepts being significantly smaller (low $l_1$ norm). We hypothesize that the information in many adjective and verbs can actually be encoded into the noun itself, resulting in this phenomenon. For example, the concept "lemon" is a more succinct form of "yellow" and "fruit".

## B.14 Experiments on Alternative CLIP Architecture

We present cosine reconstruction and zero-shot accuracy experiments with an alternative CLIP architecture from OpenAI with a ResNet50 backbone for the vision encoder. Note that these experiments were done with a 10000 size vocabulary of only one-word concepts. We find that results are similar to those presented in 3, save for OpenAI's ResNet50 CLIP performing much worse than OpenCLIP's ViT B/32 backbone in general.

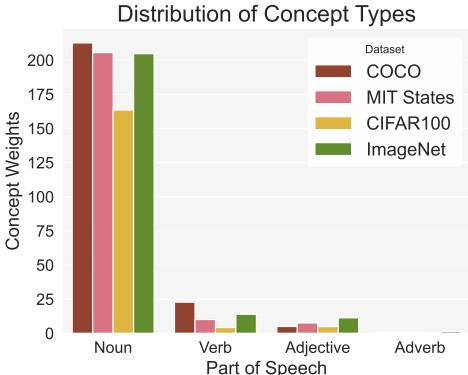

Figure 14: SpLiCE decompositions are mostly comprised of nouns across multiple datasets.

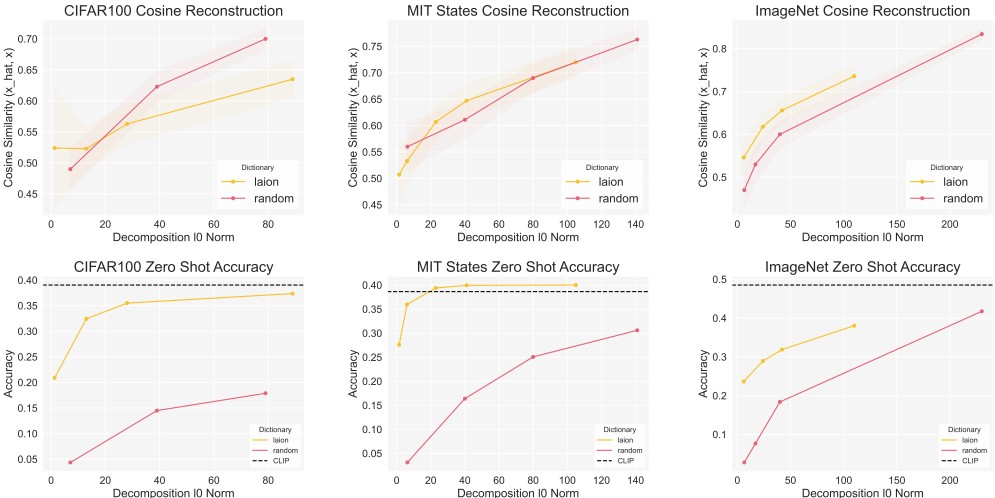

Figure 15: Performance of SpLiCE decomposition representations on zero-shot classification tasks (bottom row) and cosine similarity between CLIP embeddings and SpLiCE embeddings (top row) for OpenAI's ResNet50 CLIP model.

