# OpenReview forum: "Interpreting CLIP with Sparse Linear Concept Embeddings (SpLiCE)"
_NeurIPS.cc/2024/Conference — NeurIPS 2024 poster_

### Official Review · Reviewer_djnP · 2024-06-25

**Soundness:** 3
**Presentation:** 4
**Contribution:** 2
**Rating:** 7
**Confidence:** 4

**Summary:**

Whilst useful for many downstream tasks, CLIP’s vision-language representations are notoriously hard to interpret. The paper proposes to represent CLIP representations in a sparse, non-negative overcomplete basis of learnt interpretable directions using a standard dictionary learning technique. Not only is interpretability increased, but the core claim is that this does not harm the performance of the downstream applications significantly. Such a model would provide interpretability “for free”, in the sense that practitioners could opt for the interpretable version without any performance cost (which would constitute a significant achievement).

---

Post-rebuttal:
Following a productive discussion with the authors (who both clarified many experimental details and proposed revisions to the claims of the paper to better reflect its results), I think the paper will be of great value to the community and recommend its acceptance.

**Strengths:**

- The work strives to address a vitally important task: to design models that offer interpretability at no cost to performance. Such an endeavor is particularly important practically in allowing practitioners to adopt interpretable models in practice (by not trading off performance).
- The paper is very well-written, the applied methodology is appropriate, and the focus on CLIP representations means the paper has potentially great relevance and reach to practitioners.
- Offering more than just transparency, I appreciate the method's multiple use cases--not only for discovering spurious correlations but also for model editing.

**Weaknesses:**

# [W1] Sparsity/interpretability does trade-off accuracy

I was very excited about the paper’s bold claim in the introduction to `provide interpretability, at no cost to downstream performance` ([L6], but also at [L349]) upon a first read. Unfortunately, from what I came to understand later from the results of the proposed dictionary in yellow plots in Figure 3, this claim appears false.

There *is* a cost to zero-shot classification performance (which is arguably one of the most important downstream applications of CLIP). For example, small L0 norms lead to as much as ~10% accuracy drop on CIFAR100 (left-most plot), and what appears to be a ~20% performance drop on ImageNET1k using the raw dictionaries. A similar pattern of performance degradation for high sparsity is observed in Fig. 7 for retrieval tasks. Thus, it is simply not true that the method is at “no cost” to performance, and arguably it is not even at “minimal” cost (used later in [L52]). These results unfortunately do not support the paper’s core claims.

Crucially, if I correctly understand the x-axis as quantifying the number of non-zero coefficients, this is even more problematic. High sparsity is desirable for interpretability [L164]—but we see from the experiments that too small an L0 term trades-off accuracy, and sometimes rather significantly so. This fundamental trade-off is a clear critical limitation of the work (in conflict with the paper's core claims) and puts interpretability and accuracy at odds—but it is not ever stated or discussed as a key limitation of the work.

The authors should revise this core claim in light of the fact there is indeed a cost to performance increasing as a function of sparsity, and include a dedicated discussion of such limitation. Furthermore, I would expect to see many more experiments on the standard zero-shot datasets used in the original CLIP paper [1] to evaluate this core claim, not just on 3 (given this is zero-shot, these experiments are very fast to run with pre-trained dictionaries).

This weakness in particular swayed my rating of this paper negatively, and I am happy to consider revising my score accordingly, should an adequate answer be provided in response.

# [W2] Limited new technical contributions, but also lacking technical insights/comparisons to alternative existing solutions

The paragraph in [L229] is rather brief and states that sklean’s Lasso solver is used. In the concrete application of sparse dictionary learning for CLIP representations in the paper, a representative reader would be very interested to see a much richer exploration of alternative approaches to solving the model objective.

For example, in CLIP, how does this proposed solution compare to non-negative KSVD? How about NMF (with a sparsity penalty)? Or sparse autoencoders? Or learning this end-to-end with projected gradient descent (to project the coefficients onto the non-negative orthant), etc.

Given that the paper does not offer any novel technical contributions (which is okay!), I think the paper would have been much stronger if at least some technical insights (about existing techniques) as they pertain to representation learning with CLIP were provided. At a minimum, a *discussion* of some of the other techniques listed above (and why they might not be appropriate) would be desirable, and experiments comparing the results from each to validate the proposed solution would make the paper’s contributions even stronger through technical insight.

# [W3] Unnecessary mathematizing (minor)

As a reader very excited about this paper’s application, I feel the second half of Section 3. “When sparse decompositions exist” is an unnecessary formalization ([L141] onwards). I’m unconvinced of the need to introduce “Proposition 1”—at minimum, it feels like a distraction from the other interesting results in the paper. As far as I can see, this proposition is not used at all in the paper, and no experiments are conducted to show the 5 assumptions are reasonable (aside from Assumption 3).

The “assumptions” 1-5 listed do appear useful heuristics to reason about when Sparse decompositions are appropriate. As the authors also state [L133], concept linearity seems most crucial here. As a reader, what I care about is how these assumptions hold in CLIP in practice. As such, it would be much more useful to see instead the key experiment in Appendix B.5 discussed here in the main paper to support this critical assumption.

To summarize this final weakness: my impression is that the paper’s clarity could be improved by (a) considering deferring the formalizations to the supplementary material, or reducing their significance/length in the main paper and (b) considering making the critical experiments validating the key assumption in Appendix B.5 more prominent. I believe the paper would be much more impactful and have greater reach if the key results were better highlighted, and math that is not strictly necessary dropped.

---

* [1]: Radford, Alec et al. “Learning Transferable Visual Models From Natural Language Supervision.” International Conference on Machine Learning (2021).

**Questions:**

# [Q1] Vocabulary choice and dataset dependence

I’m not sure I would agree with the claim that the `efficacy of the decomposition is (in principle) independent of individual datasets`. Isn’t the vocabulary choice made in this work specific to LAION-400m? Perhaps using the WordNET nouns/adjectives as the vocabulary (and bigrams formed by common combinations) would be even more dataset/task agnostic?

# [Q2] Mean centering

As stated by the authors, the CLIP representations live on a hypersphere. Instead of taking the geometric mean in Euclidean space (as part of addressing the modality gap), did the authors explore taking the Frechet mean? This seems like it would better account for the geodesic distances between the points.

# [Q3] Zero-shot performance in the limit
What happens to the zero-shot performance in Fig. 5 in the limit of the l0 norm being equal to the dimensionality of the space? Shouldn’t we expect matching performance to the CLIP baseline? It would be insightful to extend the x-axis to observe how it performs in the limit.

**Limitations:**

No, limitations are not adequated addressed, and the current limited discussion should be placed prominently in the main paper rather than the supplementary material.

---

> ### Author Rebuttal · Authors · 2024-08-07
>
> Thank you for your comments! We appreciate your feedback and address your concerns below.
>
> **Sparsity/interpretability-accuracy tradeoff.** We thank the reviewer for their detailed thoughts on the interpretability-accuracy tradeoff claims and for their willingness to increase their score if this issue is resolved. The reviewer is correct that Figure 3 highlights the existence of tradeoffs between interpretability and accuracy, and we will correct the language around these claims and address this limitation in the final version. However, we note that this tradeoff is small at desirable sparsity levels. For a sparsity level of 20-30 (chosen with respect to a prior human evaluation study [49]), we find a drop in zero-shot accuracy of less than 4% for the CIFAR100 dataset (Fig. 3). Furthermore, this drop is entirely recovered when classification is done via probing instead of zero-shot (please see Appendix C.1). Secondly, this tradeoff is not necessarily fundamental, as it is possible to retain downstream performance by using SPLiCE as a post-hoc concept explanation method. More specifically, we can add the residual error back into the reconstructed embedding to recreate the CLIP embedding before use in downstream applications (as suggested by reviewer Vxod). Thus, we can achieve interpretability without affecting performance. We do note that in this case, the residual will still remain uninterpretable, which is why we do not focus on this setting in the main paper. We will be sure to discuss this in more depth in the final version, including the key limitations of both of these settings.
>
> As requested by the reviewer, we included experiments on four additional datasets from the CLIP paper [1] (Caltech-101, SUN397, STL10, VOC 2007) in the additional results, Table 1, for sparsities of 20-35. While there is a decrease in performance, we believe it to be relatively small and recoverable by probing or the technique discussed above.
>
> **Technical contributions and exploration of alternative existing solutions.** Thank you for this interesting question. We note that we explored two alternative approaches in the main paper, including an alternative solver, ADMM, and a method for learning the dictionary, FISTA, but we will elaborate on this discussion further in the final version. In Appendix B.2, we discuss the use of Alternating Direction Method of Multipliers (ADMM) in place of scikit-learn's Lasso method to solve the SPLiCE objective with GPU/batched speedup. We present results in Figure 2 of the additional material, illustrating the equivalence of these methods in both cosine reconstruction and zero-shot accuracy for CIFAR100.
>
> We also explore an alternative dictionary learning method, similar to those suggested by the reviewer (Sparse NMF, KSVD, Sparse Autoencoders, PGD), of sparse dictionary learning with nonnegative weight projection via Fast Iterative Shrinkage-Thresholding (FISTA) in Figures 3, 7. We note that FISTA and the other suggested methods all learn uninterpretable dictionaries that require post-hoc human labeling of the learned concept dictionary atoms. While existing literature such as [31] explores methods for visualizing concepts learned by NMF, PCA, k-Means, and Sparse Autoencoders, this process still requires manual labeling of concepts, which can be both data- and time-intensive. One of the key benefits of SpLiCE is that we fix the concept atoms a priori rather than having to analyze and label components post-hoc. Our results also demonstrate that a learned dictionary results in better reconstruction in terms of cosine similarity, as expected, but surprisingly underperforms our LAION dictionary on zero-shot classification, due to the lack of semantic structure within these learned dictionaries.
>
> **Mathematizing.** Thank you for the comment. We plan to adopt your feedback, move the empirical validation found in Appendix B.5 to the main paper, and reduce the length of our formal analysis, deferring our theoretical results to the appendix. For further discussion, please see our general comment.
>
> **Vocabulary choice.** This is an important point we intend to discuss in the final paper. While we considered WordNet, we preferred LAION as our source of concepts because it is the training dataset of Open-CLIP (the model used in this work), preventing us from including concepts in the dictionary that CLIP may not have learnt. Please see our general comment for a more thorough discussion of this point.
>
> **Mean centering.** We did consider Frechet means, but we found it non-trivial to define a clear notion of mean-centering on the hypersphere. Because the surface of a unit sphere does not contain the “zero” element necessary to define an algebraic field, it is hard to define addition or subtraction on the sphere and, subsequently, mean-centering. We found that Euclidean mean-centering sufficiently closed the modality gap for our purposes; however, if the reviewer has specific suggestions regarding this, we would be happy to update our centering method.
>
> **Zero shot in the limit.** We do expect to recover CLIP’s baseline performance in the limit when our sparsity constraint allows for a 512-dimensional solution, aside from shrinkage incurred by the L1 penalty. In our experiments, we find that sklearn’s Lasso algorithm does not converge with minimal L1 regularization, so we solve with Ridge regression and truncate to the top 512 coefficients. Table 2 in the additional material shows that at 512 coefficients we are indeed able to recover CLIP’s baseline performance. Note that this still provides interpretable decompositions  and so is directly preferable over CLIP.
>
> **Limitations.** We will be sure to move our limitations section up to the main body of the paper for our final submission and will elaborate on the accuracy-interpretability tradeoff you have mentioned above.
>
> We again thank you for your feedback and hope these comments will encourage you to reconsider your score.

---

> > ### Comment · Reviewer_djnP · 2024-08-08
> > **Thanks to the authors; remaining question about probing in C.1**
> >
> > Thanks to the authors for their thorough reply! I appreciate the authors’ extra results, technical discussions, and re-organization of the paper based on the comments.
> >
> > I like this paper and appreciate its goals. To be clear, I think it’s reasonable to expect such a performance degradation for high sparsity levels. But the fact this happens should be made crystal clear, or else the paper runs the risk of being inadvertently misleading through the current claims in the abstract (and main paper). Some responses to the authors’ rebuttal:
> >
> > **Zero-shot performance**
> >
> > It is indeed helpful to interpret the results of Figure 3 again with the understanding that a sparsity level of below 32 is preferable according to human studies—similar to how it already appears in the discussion of related work, it would be insightful to include a brief discussion of this previous papers’ result to help interpret Figure 3 inline.
> >
> > I would respectfully disagree with the authors about the significance of the accuracy drop however: ~30 is near the upper bound of desirable levels of sparsity according to [49]. But this already brings a 4% drop in accuracy on a dataset as simple as cifar100--this seems to me a rather notable degradation. I don’t think one can use the word “small” to describe this.
> >
> > **Probing experiments**
> >
> > I do not quite understand the experimental setup in Appendix C.1. Wouldn’t “retaining performance” of CLIP mean that we match the performance of a linear probe trained on the CLIP embeddings’ training set and also evaluated on the CLIP embeddings’ test set? These are the capabilities we wish to retain.
> >
> > **Re-adding the residual error**
> >
> > This is indeed a smart idea for recovering performance. As the authors state though, the residual vector will remain uninterpretable. Therefore, I’m not convinced that this is a viable argument for the method offering interpretability whilst retaining performance.

---

> > > ### Author Response · Authors · 2024-08-09
> > > **Discussion of remaining questions**
> > >
> > > Thank you for taking the time to discuss our work! We sincerely appreciate your feedback and the improvements you have made to this work.
> > >
> > > **Accuracy Interpretability Tradeoff.** We completely agree with your sentiment that communicating the accuracy-interpretability tradeoff needs to be crystal clear. To address this, we attach below the exact instances of how we will change the wording of our paper to avoid misleading readers.
> > >
> > > In addition, we will move the limitations section up to the main paper and include the following. “We note that SpLiCE, similar to many other interpretability methods, presents a tradeoff between interpretability and performance. Previous work has shown that sparsity is necessary for human interpretability, but sparsity also results in information loss. While this work aims to limit such loss and provide users with the ability to choose what is suitable for their application, addressing this tradeoff still remains an open issue, and we hope that future work can build upon ours to address this.” Please let us know if this is satisfactory.
> > >
> > > **Zero-shot.** In our results section 5.2, we can elaborate on the results of [49], explaining that “Ramaswamy et al. [49] conducted extensive user studies and found that most participants prefer up to 32 concepts in terms of catering to human preferences, by which point we can see the reconstructed embeddings have recovered significant performance.” Furthermore, in our figures, we can include a vertical dashed line in the final version indicating this human preference.
> > >
> > > We agree for a simple dataset like CIFAR100 a 4% drop in accuracy may not be considered small to some readers. However, given the massive gain in interpretability offered by SpLiCE embeddings, others may view this trade-off as worth the cost. Furthermore, we offer a Pareto frontier of interpretability and accuracy options as a function of our sparsity penalty in our SpLiCE decomposition (Fig. 3). If a reader finds 4% to be unacceptable, they can decompose a lower penalty, resulting in a more accurate reconstruction and improved downstream performance. We acknowledge that this may contradict our prior point that a prescribed sparsity of 32 is ideal for human interpretability, but our main idea here is that a user can choose what level of interpretability and performance is suitable for their needs. We also note we include experiments across 7 datasets including those in the additional material showing similar results on varied and more complex datasets.
> > >
> > > **Probing.** Our apologies for the confusion, we will clarify the experiment here. In Table 3 (CIFAR100) and 4 (MIT States), we train two probes, either on the SpLiCE reconstructed embeddings (top row) or the original CLIP embeddings (bottom row). Then, we test these two probes on SpLiCE reconstructed embeddings of various sparsities (left 4 columns for CIFAR, left 3 columns for MIT States) and also the original CLIP embeddings (rightmost column). Our baseline is the CLIP probe trained and evaluated on CLIP embeddings (bottom row, rightmost column), and our results show that when probing, SpLiCE decompositions preserve the performance of the CLIP probe (bottom row) and training a probe on SpLiCE reconstructions results in a minor performance drop when compared to training a probe on CLIP itself. We hope this clears up any confusion!
> > >
> > > **Re-adding the residual.** We agree that this setup is not ideal for maintaining full interpretability, and thus we do not include it in our method and experiment design. We simply wanted to bring it up as an alternative if performance is of utmost importance to a user. We are happy to leave this out of the paper if desired.
> > >
> > > Thank you again for all your comments and suggestions! We hope you find our responses satisfactory and will consider increasing your score. If not, please let us know of any remaining concerns and  we will be happy to continue discussing.

---

> > > > ### Author Response · Authors · 2024-08-09
> > > > **Proposed Rewording**
> > > >
> > > > Here are our proposed changes to the wording of our work:
> > > >  * Abstract, Lines 6-7. “...can be leveraged to provide interpretability, at no cost to downstream performance, by decomposing representations into semantic concepts” >> “...can be leveraged to provide interpretability, by decomposing representations into semantic concepts”
> > > >  * Intro, Lines 45-46. “Remarkably, these interpretable SpLiCE embeddings perform comparably to black-box CLIP representations on metrics such as zero-shot accuracy” >> “Remarkably, these interpretable SpLiCE embeddings have favorable accuracy-interpretability tradeoffs when compared to black-box CLIP representations on metrics such as zero-shot accuracy”
> > > >  * Intro, Line 52: “representations with minimal loss in performance on downstream tasks” >> “representations with high performance on downstream tasks”
> > > >  * Results, Lines 248-249: “ SpLiCE closely approximates the performance of the black-box CLIP representations” >> “SpLiCE efficiently navigates the interpretability-accuracy pareto frontier and retains much of the performance of black-box CLIP representations”
> > > >  * Conclusion, Lines 349-250: “the improved interpretability of SpLiCE does not come at a cost to downstream performance” >> “the improved interpretability of SpLiCE allows for users to diagnose and fix model mistakes, ideally increasing the effectiveness and performance of the overall system they are using CLIP in. Furthermore, SpLiCE allows for an adjustable tradeoff on the interpretability-accuracy pareto frontier, enabling users to decide the loss in performance they are willing to incur for interpretability.”

---

> > > > > ### Comment · Reviewer_djnP · 2024-08-10
> > > > > **Thanks for the great discussion!**
> > > > >
> > > > > A big thanks to the authors for their ongoing efforts in this exchange.
> > > > >
> > > > > Indeed, the probing experimental setup is now a lot clearer in my mind (thanks for this clarification!), and I appreciate your making the proposed revisions to the claims explicit. I think "SpLiCE as well-navigating the interpretability-accuracy pareto frontier" is a great characterization of the paper and its results.
> > > > >
> > > > > Thanks again to the authors for the respectful and productive discussion--I've increased my support of the paper, and I believe it would be of great value to be presented at the conference.

---

> > > > > > ### Author Response · Authors · 2024-08-13
> > > > > > **Thank you!**
> > > > > >
> > > > > > We sincerely appreciate your decision to raise your score after reviewing our rebuttal. Your feedback and engagement throughout the review and discussion process has improved our paper, and we will be sure to incorporate these changes into our final version. If you have any further questions or concerns, please feel free to discuss with us.

---

### Official Review · Reviewer_nCHL · 2024-07-14

**Soundness:** 2
**Presentation:** 3
**Contribution:** 2
**Rating:** 6
**Confidence:** 3

**Summary:**

This paper introduces a method for building sparse embeddings from CLIP in order to improve the interpretability of CLIP’s latent space.  They formulate their objective as a sparse reconstruction problem, and under certain assumptions, demonstrate when finding a sparse representation is possible.  Empirically, the authors show over three datasets that zero-shot accuracy and reconstruction errors are correlated, and performance is similar with sparse embeddings vs. without.  They further illustrate the benefits of sparse embeddings for interpretability noting applications to reducing spurious correlation reliance on a Celeb-A benchmark, and identifying biases through representations.

**Strengths:**

- This paper presents an important study toward understanding the latent space of vision-language models like CLIP which has become a very popular paradigm and backbone for many downstream tasks. This can help to improve the next generation of such models in training and downstream use. The use of text to interpret to solely understand the vision space and perform interventions is also important and makes the method relatively lightweight.  It also makes the method extendable to researchers and practitioners outside of the space as only concept vocabulary is needed.

- The paper is overall well-written stating clearly the assumptions of the work in Section 3, and the proposed method is also clear and intuitive to implement.

**Weaknesses:**

- The primary weakness of this paper is that it is unclear to me what improvements come from generating sparse CLIP embeddings.  Results indicate that there is no improvement in zero-shot accuracy due to some outstanding reconstruction error.  In contrast, when removing spurious correlations it is typically seen as an improvement to model performance as the model would typically in this case then be evaluated in the OOD setting where the correlation is not present.  From this perspective, I think the authors might benefit from changing to a setting more similar to this, where the results are more aligned with benefits from having an interpretable tunable mechanism, rather than the more ID zero-shot performance settings.  This is started in the model editing, however it doesn’t lead to much improvement from correlation reliance, only mitigation of the capability to classify another attribute.  In contrast performance goes down indicating some capability but perhaps not the right evaluation setting.

- Similarly, I am uncertain about the case study evaluations for discovering spurious correlations.  Results for the man-woman evaluation appear to only indicate the trend is present in 70/600.  This seems like a relatively low number as the majority then do not have this bias.  It is unclear what the impact would be in terms of downstream performance for classification on CIFAR-100, or a gender classifier trained on CLIP embeddings.  The study seems inconclusive due to the results not being well contextualized 70 out of 600, and the lack of a downstream performance evaluation and improvement.

- Finally, in multiple places, the authors mention improved human interpretability such as 244, 277, etc. however the authors never conduct any human evaluations.  If the authors claim improved human interpretability, the authors should conduct such a human evaluation, otherwise it is only hypothesized improved human interpretability.

**Questions:**

Have the authors demonstrated an evaluation setting where having the more interpretable concept features would improve performance? I believe some OOD setting such as spurious correlation could improve, but there may be many setting this is worse, as you are reliant on the concept features and may be worse for settings such as fine-grained classification where the concepts were not covered.

**Limitations:**

- An additional limitation of the work not discussed by the authors is the reliance on writing a set list of word-level concepts.  The approach is reliant on this set list having the desired concept when detecting spurious correlations, etc.  As the authors have pointed out this also limits the type of concepts.

- The authors have also not discussed how practical the assumptions proposed in Section 3 are, and whether they are correct in practice.  This is important for demonstrating applicability of the proposed work.

---

> ### Author Rebuttal · Authors · 2024-08-07
>
> Thank you for your comments! We appreciate your feedback and address your concerns below.
>
> **Benefits of sparse CLIP embeddings.** Thank you for highlighting this confusion, we hope to clarify this in our response. The primary benefit of SpLiCE is the insight it provides into interpreting the semantic content of the underlying images encoded by ordinary CLIP embeddings. We show in the paper (in Section 6) that this additional insight opens up at least two applications: detecting biases in datasets and intervention on the concepts encoded, both of which are not possible with ordinary CLIP representations. SpLiCE provides a computationally efficient method for understanding the semantics of individual images (See Fig. 2 and additional results Fig 3) and for understanding the semantics of unstructured datasets, allowing for the exploration and summarization of data even without labels. We answer the specific questions that the reviewer has about the two applications below.
>
> **OOD evaluation setting.** Thank you for bringing up this important evaluation setting. We kindly note that we already consider the OOD setting in the Appendix C.5 with the Waterbirds dataset, which has a spurious correlation in its train set and a distribution shift in its test set. SpLiCE allows us to identify distribution shifts in two ways: first, we can evaluate the similarity in distributions between decompositions of test images and validation images (as we did in Table 6 of the Appendix for Waterbirds), or we can assess the weights learned by a linear probe to better understand how correlations in the train and test data might differ. For example, we see that we can detect the spurious correlation of landbirds and land backgrounds in our probe trained on SpLiCE weights, and when intervening on the highly-weighted land-related concepts [“bamboo”, “forest”, “hiking”, “rainforest”, and any bigrams containing “forest”] by setting them to zero in the weights of our probe, we are able to increase accuracy for the subgroup “waterbirds on land” in the test distribution (Appendix Table 5). In other words, SpLiCE’s interpretable sparse decomposition admits the identification of OOD spurious correlations (e.g., landbirds and land backgrounds), and also provides an interpretable strategy for intervention by allowing for fine-grained control via semantic concepts.
>
> **Significance of Woman-Swimwear bias.** Thank you for the question. This study shows that within the “swimwear” attribute, women are significantly overrepresented when compared to men. Another way to view this is that 10% of the images of women in CIFAR100 depict them in swimwear (generally referring to bikinis) or underwear. We believe this is a serious problem of bias in the dataset, and constitutes a “representational harm,” in that these stereotypes present in the dataset will be propagated with the use of this dataset to downstream tasks. This can result in downstream risks of bias such as 1) a generative model learns that when prompted for photos of women, it should generate this woman in revealing clothing or underwear, or 2) an image retrieval or generation algorithm failing to be representative when queried for “A photo of a CEO” by returning primarily men due to women being less likely to be depicted in corporate attire. While the reviewer points out the lack of downstream performance evaluation and improvement, we highlight that the goal of our work and this case study is not to develop a bias intervention but rather to propose an interpretability method and demonstrate its usefulness, and that this bias is a property of the dataset itself, which can result in different impacts for different models and tasks.
>
> **Human evaluations.** We agree that user studies are an essential part of evaluating human interpretability, and we have thus included results for a small-scale human evaluation based on the user study presented in [23] in our additional results (Figure 1). We evaluate our method along with two baselines [22, 23] to assess how relevant the concept decompositions are to the input images, how relevant the concept decompositions are to the model’s prediction, and how informative the decompositions are. We find that users significantly prefer our method to the baselines for both relevance to the input images as well as informativeness, further validating the interpretability of our method. We direct the reviewer to our general comment for more information about the study setup and results.
>
> **Reliance on set list of concepts.** Thank you for this comment. We agree that spurious correlation and bias detection are dependent on the spurious features being present in the concept vocabulary, thus posing a limitation. We note that we attempt to make our vocabulary as general as possible by constructing it from LAION, the training dataset of Open-CLIP, however it is true that this set may still not include the desired concepts for some downstream applications. In addition, we note that we find this vocabulary to be empirically sufficient for a variety of tasks (For more tasks see additional results). However, we will be sure to acknowledge this limitation in the paper and make this point more clear in the final version. For a more detailed discussion on our constructed vocabulary, please see our general comment.
>
> **Practicality of Assumptions.** This is an insightful question. Our empirical evaluations all strive to validate the assumptions proposed in section 3. In addition, we include in Appendix B.5 an additional sanity check of CLIP’s linearity (Assumption 3). We note that we intend to de-emphasize these assumptions in the final version of the paper in favor of our empirical validation from Appendix B.5, as suggested by reviewer djnP. For further discussion of these assumptions, please refer to our general comment.
>
> We again thank you for your feedback and hope these comments will encourage you to reconsider your score.

---

> > ### Comment · Reviewer_nCHL · 2024-08-12
> > **Reviewer Response to Author Rebuttal**
> >
> > Thank you for the comments and addressing many of my concerns.  My primary concerns with the paper were the claims that (1) the representations and concepts were human interpretable, and (2) there was no decrease in performance.  I see that concern (1) has been addressed, however I believe (2) has not been addressed sufficiently in the rebuttal and is still a clear limitation of the proposed work.
> >
> > Nonetheless, I am increasing my score following the addition of (1) in the rebuttal pdf as well as the desired inclusion of a more thorough study in the final version.  Regarding (2), I am satisfied with the wording proposed by the authors in Review djnP.  I believe authors should also address this wording in the Abstract of the paper as well: "In this work, we show that the semantic structure of CLIP's latent space can be leveraged to provide interpretability, at **no cost** to downstream performance, by decomposing representations into semantic concepts." to reflect that there is at least a "small" cost.

---

> ### Comment · Area_Chair_ZzCV · 2024-08-12
> **Gentle reminder to reply to rebuttal**
>
> Dear Reviewer nCHL,
>
> As the author-reviewer discussion period is about to close, I would like to know your thoughts on the author rebuttal.  Especially since the other reviewers all currently leans towards acceptance, it would be extremely informative for me to know if you still maintain your original score following the rebuttal.  I would very much appreciate if you could reply to the authors before the close of the discussion (Nov 13 11:59 pm AoE).
>
> Gratefully,
>
> AC

---

> > ### Author Response · Authors · 2024-08-13
> > **Thank you!**
> >
> > We sincerely appreciate your decision to raise your score after reviewing our rebuttal. Your feedback during the review process has improved our paper, and we will be sure to incorporate these changes into our final version. If you have any further questions or concerns, please feel free to discuss with us.

---

### Official Review · Reviewer_jbkR · 2024-07-18

**Soundness:** 3
**Presentation:** 3
**Contribution:** 3
**Rating:** 6
**Confidence:** 4

**Summary:**

This paper presents a method to explore semantic concepts in multimodal models of text and images. Specifically, the paper formulate semantic concept discovery  problem as one of sparse recovery and build a novel method, Sparse Linear Concept Embeddings (SpLiCE), for transforming CLIP representations into sparse linear combinations of human-interpretable concepts.

**Strengths:**

1. The paper is well written and reads well.

2. The technical method is generally make sense .

3. The contribution is enough for covering text-image multi-model method to an interpretable model, and the research topic is worth pursuing.

4. Sufficient experiments demonstrate the effectiveness of the proposed simple method.

5. The experiment designed in the paper is make sense.

**Weaknesses:**

see questions

**Questions:**

1 The connection and difference between your method with a multi model topic model? In my opinion, topic model is good at extracting interpretable concepts.

2 Although a lot of experiments were conducted in the paper, I think how to judge the concept discovery results is still not convincing enough. Whether to add manual evaluation experiments will be more convincing.

3. Why does negative concept weight perform better in Table 5? What does this mean?

**Limitations:**

see weakness and question

---

> ### Author Rebuttal · Authors · 2024-08-07
>
> Thank you for your comments! We appreciate your feedback and address your concerns below.
>
> **Connection to Multi Modal Topic Models.** This is an interesting connection! MMTMs such as [A] are trained on a corpus of data and use tf-idf statistics to generate multimodal clusters of topics, such as clustering images with text topics. However, these models are designed to explain datasets and not individual samples like SpLiCE. Additionally, this means every image in a cluster gets the exact same image tag and score, which prevents any comparison or differentiation between samples in a dataset despite these images being unique. Furthermore, we could use SpLiCE as a MMTM by generating concept decompositions for each image and applying simple k-means clustering on our interpretable decompositions. Finally, we note that the goals of this work are not to explain a dataset but rather to study CLIP and leverage it to explain individual image embeddings. In doing so we unlock new use cases, one of which is the capability to explain a dataset. We thank you for pointing out this connection, and we will be sure to discuss this work in the final version of our paper.
>
> **Manual evaluation experiments.** We agree that human evaluation studies are important for assessing the human interpretability of concept based explanations. We have included results from a small-scale human evaluation based on the user study presented in [22] in our additional results (Figure 1). We evaluate our method along with two baselines [22, 23] to assess how relevant the concept decompositions are to the input images, how relevant the concept decompositions are to the model’s prediction, and how informative the decompositions are. We find that users significantly prefer our method to the baselines for both relevance to the input images as well as informativeness, further validating the interpretability of our method. Please see our general comment for more information, where we elaborate on the setup and results of this evaluation.
>
>
> **Negative concept weight performance.** Thank you for the observation. As part of our desiderata for interpretability, we impose a nonnegativity constraint on the concept weights, as user studies (including our own) highlight that humans find negative concepts and weights to be confusing and unintuitive. In our optimization problem (2), we try to find a set of weights that minimize our reconstruction error. Constraining this set of weights to be nonnegative will reduce the search space and thus result in a worse reconstruction. So, we see in Table 5 that removing the nonnegativity constraint results in a more accurate reconstruction, but we lose the interpretability of our decompositions.
>
> We again thank you for your constructive feedback!
>
> [A] Grootendorst, M. (2022). BERTopic: Neural topic modeling with a class-based TF-IDF procedure.

---

### Official Review · Reviewer_acbD · 2024-07-26

**Soundness:** 2
**Presentation:** 3
**Contribution:** 3
**Rating:** 6
**Confidence:** 3

**Summary:**

This paper introduces a method to transform CLIP representations into sparse linear concept embeddings that are interpretable to humans. SpLiCE uses task-agnostic concept sets, demonstrating its versatility over prior works. SpLiCE provides interpretability without compromising zero-shot classification performance and shows further applications, including spurious correlation detection and model editing.

**Strengths:**

SpLiCE suggests a novel method to interpret CLIP embeddings into semantic concepts, which could be a good way to understand the latent space of CLIP. As it is a task-agnostic approach, which does not constrain itself to a certain domain, it can be applied to various datasets to explore the CLIP embeddings. This task-agnostic concept shows its scalability across different datasets, including CIFAR100, MIT States, CelebA, MSCOCO, and ImageNetVal. Also, the authors demonstrate the efficacy of their method both quantitatively and qualitatively.

**Weaknesses:**

[W1] I believe the choice of datasets is insufficient to show the efficacy of "task-agnostic" concpet sets. For example, how does SpLiCE work on bird identification dataset or CelebA? What are the top activated concepts in these datasets?

[W2] Adding class labels to concept dictionary seems inappropriate from the perspecive of "concept decomposition," as in the case of ImageNet. If the top concept is class label itself, what is the need for concept decomposition?

[W3] A user study on the interpretability of SpLiCE would be needed to quantify how interpretable this method is, as in [23].

[Miscellaneous] I think line 96's citation [22] should be changed to [23].

**Questions:**

[Q1] Does the concept decomposition differ between correct and wrong samples (from the perspective of zero-shot classification) ?

[Q2] As the effectiveness of SpLiCE heavily relies on the quality and comprehensiveness of the concept dictionary derived from the LAION dataset, is there a possibility that limitations or biases in this dictionary could directly impact the performance and interpretability of the method?

**Limitations:**

Please refer to the weaknesses and questions part.

---

> ### Author Rebuttal · Authors · 2024-08-07
>
> Thank you for your comments! We appreciate your helpful feedback and address your concerns below. We will also correct the typo you mentioned.
>
> **Efficacy of task-agnostic concept sets.** As requested by the reviewer, we include in the additional results concept decompositions of randomly chosen samples from Waterbirds and CelebA (Fig. 3b). We find that the decompositions include fine-grained and detailed concepts, such as the celebrity “Rihanna”, the brand “adidas” (in reference to a branded Adidas sweatband), and the species “pelican”. This shows qualitatively that our concept set is indeed task agnostic or at least broadly applicable to a variety of tasks. While we only include four in the additional rebuttal documents due to space constraints, we will include a more thorough set of examples in the final version. We also include quantitative results for a broader set of tasks from the original CLIP paper [1] in our Additional Materials (Table 1), as suggested by reviewer djnP. These results further indicate the wide applicability of our selected concept set. If the reviewer believes that the term “task-agnostic” is still inappropriate for this method, we are happy to change our language accordingly.
>
> **Adding class labels to the concept set.** Thank you for this comment. For ImageNet, we find that the class names are often fine-grained animal species that are more than two words long (such as “European fire salamander”, “sulphur-crested cockatoo”, “red-backed sandpiper”, etc), making it difficult for one- and two-word concepts to capture. As such, these concepts can be added to the LAION dictionary to allow for full reconstruction, while maintaining the interpretability of all other concepts in the image. We include this augmented vocabulary in our experiments to demonstrate the efficacy of our sparse decomposition method, even if it highlights a limitation of the LAION vocabulary.
>
> We also note that many images, including those in ImageNet, can be complex and contain semantic information outside of the class object. Even if the top concept is the class label, the  rest of the decomposition may contain information relevant for prediction, especially if there are spurious correlations. Decompositions should include all of this information to allow for full understanding of the semantic content of the image, and more importantly to allow for use cases outside of prediction, such as editing. If you take the example of the Waterbirds dataset, we expect our decompositions to include both the class label and the land/water background concepts. Being able to edit the background concepts while maintaining the class label is useful for improved classification. Furthermore, this semantic information can be useful for other applications, such as dataset summarization, image tagging for retrieval, and exploration of unlabelled data. In summary, concept decompositions that include a combination of class concepts and other semantic information provide utility in a variety of settings beyond prediction; however, we will further clarify this in our final paper.
>
> **User study.** We agree that user studies are important for assessing the human interpretability of concept based explanations. We have included results from a small-scale user study similar to that suggested by the reviewer in [23] (Task 2) in Figure 1 of the additional materials, where we present users with an image and two concept decompositions/explanations and ask the following questions:
>  * Which explanation is more relevant to the provided image?
>  * Which explanation is more relevant to the model’s prediction?
>  * Which explanation is more informative?
>
> We benchmark against two other CLIP-interpretability methods: LF-CBMs [23] and IP-OMP [22]. We find that users significantly preferred SpLiCE to LF-CBMs for both (1) and (3), and significantly preferred SpLiCE to IP-OMP for (1, 2, 3). We also highlight that our method is able to produce similar/better concept decompositions, in terms of human interpretability, than the baselines without the need for class labels for concept mining and without training a classification probe, both of which are computationally expensive. For more discussion, please see the general rebuttal.
>
> **Decomposition for correct/incorrect samples.** Thank you for the interesting question. In the additional material, Figure 3a, we include example SpLiCE decompositions for correctly and incorrectly classified samples using zero-shot with SpLiCE. We see that incorrect sample decompositions often contain correlated but slightly incorrect concepts that apply to other classes (“white dog” for an image of a white wolf) or concepts present in the image but not relevant to prediction (“baby toys” and “toddlers” for “abacus”). The former presents an instance where a practitioner might consider intervening on their predictor to improve performance, while the latter is simply an example of a difficult sample due to the noisiness of the image. In both cases, the explanation is useful for understanding the model’s reasoning and mistakes and can be used to intervene and improve performance.
>
> **Limitations of LAION-based concept set.** This is an important limitation of our work. It is correct that limitations and biases in the dictionary will be reflected in downstream performance. We aimed to construct our dictionary to be as comprehensive as possible by using the training set of the model itself (as LAION was used to train Open-CLIP, the model used in this paper), so we are likely to include all concepts learned by the model. We also filtered out any unsafe or NSFW samples from LAION before constructing the dataset to limit harmful content. However, we acknowledge that this is a limitation and will be sure to make this point more clear in the final version. For more discussion please see the general comment.
>
> We again thank you for your feedback and hope these comments will encourage you to reconsider your score.

---

> > ### Comment · Reviewer_acbD · 2024-08-12
> > **Great thanks to the authors!**
> >
> > I appreciate the efforts the authors have made to address my concerns! Most of my concerns are well addressed, so I'll raise my support. I hope a more thorough user study will be conducted afterward. Thanks!

---

> > > ### Author Response · Authors · 2024-08-13
> > > **Thank you!**
> > >
> > > We sincerely appreciate your decision to raise your score after reviewing our rebuttal. Your feedback and engagement throughout the review and discussion process has improved our paper, and we will be sure to incorporate these changes into our final version. If you have any further questions or concerns, please feel free to discuss with us.

---

### Official Review · Reviewer_Vxod · 2024-07-31

**Soundness:** 3
**Presentation:** 3
**Contribution:** 3
**Rating:** 7
**Confidence:** 3

**Summary:**

This paper proposes to decompose the representations of the CLIP model using dictionary learning; where the components of the dictionary are composed of human understandable concept directions. The procedure here is as follows: a concept list is constructed from a filtered set of unigrams and bigrams from the LAION-400m captions, the concept dictionary is set to be centered CLIP representations for each pre-specified concept in the list, and an optimization problem that minimizes the reconstruction loss (subject to an L-1 penalty) is specified. Solving that optimization problem results in a weight vector that indicates how the CLIP embedding can be decomposed along the pre-specified concept direction. Given this formulation, the paper solves the problem for the CLIP model and shows that it does not lead to a performance loss given thousands of concepts. In addition, the paper presents several case studies showing how the approach can be used to edit downsteam classifiers built on top of clip representations, and discover spurious correlations.

**Strengths:**

**Originality**\
This paper is the first that I know to decompose the representations of CLIP with known concepts using dictionary learning. The work is well motivated.

**Quality and Clarity**\
The paper is well-written, and clear. The related work is also discussed, and each component of the workflow presented here is well-explained and clear.

**Significance**\
This work, in my opinion, indicates that CLIP type models can be easily debugged using post hoc supervised concept learning. Overall, I think this opens up the door to more intriguing work along similar lines for larger and more complicated models.

**Weaknesses:**

Overall, I discuss some of the challenges that I see with this work. I have not much substantive qualms with this work; I just think the presentation could be tightened. However this is up to the authors to either accept or reject.

**Assumptions for the proposition not justified**:  While the assumptions made to prove proposition 1 seem natural at first glance, it is not clear to me that we can claim that they should be true. One, there is a claim that CLIP captures 'semantic' concepts and not 'non-semantic' concepts (assumption 2). In Section 5.2, the authors even confirm experimentally that the assumption is not true. Of course, the model does capture some semantic concepts, but we cannot claim that CLIP cleanly partitions these concepts as described. This is because we know of failure cases where CLIP errs precisely because it does not just capture only semantic concepts. See this paper for example (https://arxiv.org/abs/2306.12105). The failures of the CLIP model that have been observed make it clear that this assumption is not true.  Assumption 3 is somewhat problematic again because we have observed that CLIP representations indeed can be linear for *some* concepts, but it is not linear for all semantic concepts. In fact, how are we to know which concepts that CLIP considers to be semantic, and which it doesn't? Assumption 4 makes sense to me because CLIP is trained to 'align' both encoders. Assumption 5 also difficult to justify in my opinion, but I am willing to live with it. To summarize my challenge with this portion of the work, the proposition is correct, but we can't claim that CLIP behaves as described. I am bothered by these assumptions because the implication is essentially that CLIPs learns the data generating process defined here, which is not the case. This is actually a good thing because, as shown quite nicely in this work, the point of the concept dictionary 'layer' is to 'fix' the challenges with the original CLIP model.  Overall, I don't think this section of the paper can be fully justified.

**task-agnostic and without concept datasets, training, or qualitative analysis of visualizations.**:  The bolded phrase is one of the motivations for the approach described here. This is not quite true. First, you need to collect the concept dataset, the processing done with the LAION dataset is *exactly* the process of collecting a concept dataset. You do need to 'train', in the sense that you can see the process of optimizing to obtain $w$ as training. Lastly, the layer results in concept scores that can be visualized as depicted in Figure 4, so the approach here also allows us to inspect . Overall, I think this claim is hyperbolic and should be relaxed.

**Related Work and Discussion**: This work claims that it is not quite a concept bottleneck, but I disagree. It is exactly translating a concept bottleneck to CLIP. First, the methods does require annotations. To get the dictionary $C$, you need to collect a set of 10000 concepts, which are associated with particular images. The annotation process is exactly the process of manually constructing $C$. The concept list is supervised, meaning it is the authors that determine which concepts to include; of course, the process described here is not manual, but the authors should recognize that it is *still* a supervised process. The intervention procedure described in Section 6 is exactly how that is done in the CBM literature. In fact, the NMF formulation here is not new (see: CRAFT: Concept Recursive Activation FacTorization for Explainability for eg.). Overall, the approach described here is exactly what a CBM is, it is just not applied for classification. Having said all of this, I think it is fine to acknowledge the similarity.

**Questions:**

**Unsupervised or residual concepts**: I am surprised that there wasn't any significant loss in performance between the Splice representations and the original CLIP representations. Is it that 15000 concepts is enough to capture the variation? It seemed like one would want to account for the 'residual' directions in the original embedding that the reconstruction doesn't account for. Is this needed? Or do the authors think we can always get away with this kind of supervision?

**Limitations:**

Appendix A discusses some of the limitations of this work.

---

> ### Author Rebuttal · Authors · 2024-08-07
>
> Thank you for your suggestions! We appreciate your feedback on how to improve this work and address your concerns below.
>
> **Assumptions not justified.** Thank you for this comment. Regarding Assumption 2, it is true that our experiments demonstrate the presence of non-semantic concepts in our decomposition. We clarify this point further in our general rebuttal, but our intent with Section 3 and our listed assumptions was not to claim that CLIP always satisfies these assumptions, but instead to reason from first principles regarding which mathematical properties enable vision-language models to have sparse decompositions. We plan to clarify this, move our discussion of these assumptions to the Appendix, and highlight our empirical investigations of CLIP’s semantic linearity (Assumption 3, experiments in Appendix B.5.) to the main paper, as suggested by reviewer djnP. We direct the reviewer to our general comment for further discussion of this point.
>
> **“Task-agnostic and concept datasets, training, or qualitative analysis”.** Thank you for bringing this to our attention. We will be sure to update and relax our language as suggested, but we would also like to clarify our intent with this phrase here. We say that our method does not require concept datasets, in that traditional Concept Bottleneck Models (along with other methods such as tCAV) require training data that has both concept and class labels to train concept probes, whereas our method does not require a task and concept labeled image dataset. Instead, we construct our dictionary by parsing the text captions of LAION. As such, our method can be applied out of the box for a variety of tasks, without having to collect image datasets for each concept or task.
>
> Second, while we do need to optimize for our concept weights, we do not need to train any concept probes or classifiers and accordingly do not need any training data, as SpLiCE decompositions perform sufficiently in zero-shot applications. This is where our work differs from many other similar works, such as [22, 23], which require training a sparse linear layer to obtain both explanations and predictions. This also permits SpLiCE to be used post-hoc and in low-data regimes.
>
> Third, our point here was to say that we do not require qualitative analysis to label dictionary elements and thus generate explanations. More specifically, in prior work such as Sparse Autoencoders or neuron analysis in mechanistic interpretability research, dictionary elements are learned, uninterpretable vectors. As such, they require additional analysis to understand the semantics contained by each dictionary element, such as the auto-interpretation done in SAE literature, which involves decomposing large quantities of data, measuring correlations in the decompositions, and then qualitatively describing what each atom encodes. One of the benefits of SpLiCE is that we fix our concepts a priori rather than having to learn and then manually label them post-hoc which can result in errors. However, you are correct that we can use our resulting concept scores in downstream qualitative analysis by inspecting them and visualizing them.
>
> For a detailed discussion of task-agnosticity, please see our general comment.
>
> **Related work.** The connection you draw between our work and CBMs is very insightful. Our intent in claiming that we are not simply creating a concept-bottleneck model is to distinguish ourselves from works that require training classifier probes and that only consider the predictive case, as noted by the reviewer. We believe that this is a significant contribution of our work, as CLIP is frequently used in many non-predictive settings (retrieval, generation, etc), and because these decompositions allow for dataset summarization for data that is not labeled or does not have an associated task (e.g. unstructured web-scraped data). We agree that our work creates a concept bottleneck to represent CLIP embeddings, and we will discuss this further in the final version. Similarly, we agree that the vocabulary construction process is overseen by the authors, but it is not “supervised” in the traditional ML sense of the word, in that there are no labels or tasks it is optimized for. We also attempted to impose as minimal oversight and constraints on the dictionary as possible, simply ensuring that it was human-interpretable by using only 1- and 2- word concepts and safe by filtering NSFW content. Finally, we thank the reviewer for the provided citation and kindly note that we compare our work to the follow-up work by Fel et al, [31], in our submitted paper, noting that methods such as CRAFT and traditional NMF require feature visualization to understand each concept. We will be sure to clarify this comparison further.
>
>
> **Residual Concepts.** This is an interesting question and future direction! Given that our dictionary has 15000 elements in 512 dimensions, our problem is very overcomplete, so it is reasonable to assume we can reconstruct CLIP embeddings with little loss even under a sparsity constraint. One way we could account for this residual error is to add it back into our decompositions and return the original CLIP representations. However, doing so results in a component of the representations being unexplained. For example, 98% of an embedding may be explained by our sparse decomposition, but there is a risk that the 2% residual which remains unexplained contains important information for downstream tasks. Depending on the application, one may wish to incur this loss to ensure that all information remains interpretable, or they may add the residual back in for applications that require perfect performance. We find this discussion to be similar in nature to the comparisons between post-hoc explanations and inherently interpretable models, and we believe the choice should be left to the user. We will include this discussion in our final version. Thank you!

---

> > ### Comment · Reviewer_Vxod · 2024-08-12
> > **Understood**
> >
> > **Theoretical Analysis**: "we can reasonably conclude that CLIP may approximately satisfy assumptions 1-5." I think even this point about sparse decompositions is probably not true, but I have no way to refute it. I think what you have showed here is that given a pre-specified basis, then you can find a decomposition of CLIP's representation in that basis. Is one with 10k directions sparse? I don't know. Overall, I think the appeal to the linear representation hypothesis stuff is not quite clear to me. I still think even the experiments in this paper show that you can't make the claim that you seem to be making here. However, I am willing to let this go since paper reviews are public, and the community can evaluate the claim itself.
> >
> > **CRAFT et al/NMF**: I don't agree with the point that you require feature visualization for CRAFT or NMF in general. I am not sure about CRAFT, but for traditional NMF there is no explicit need for feature visualization unless I am misunderstanding what you mean by feature visualization. One could argue that Fig. 4 is exactly the type of visualization most NMF methods use, which you also use here.
> >
> > Overall, I'll be keeping my score as is.

---

> ### Author Response · Authors · 2024-08-13
> **Discussion Follow-Up**
>
> Thank you for your feedback and taking the time during the discussion period to help improve our work! We hope to answer your remaining questions below.
>
> **Theoretical Analysis.** Thank you for raising these questions regarding our theoretical analysis. As noted in the general rebuttal, we will attempt to address your concerns by moving much of the theoretical analysis section to the appendix and clearly describing its limitations and applicability in practice. We can also change our language regarding these assumptions in the Appendix and reframe them as hypotheses we have for why SpLiCE works. We appreciate your perspective on this section of the paper, and we believe your public discussion on OpenReview will be informative for future readers in the community.
>
> **CRAFT et al./NMF.** We apologize for any confusion in our response, and if we are not understanding your question correctly. Our use of the term visualization may have been unclear. We are referring to the process of interpreting concept atoms by visualizing and manually analyzing examples that activate the atom, whether generated or from a dataset. We were not referring to visualizations created with an explanation method and summarizations of explanations, such as Fig. 4 of our paper.
>
> We will focus specifically on NMF in the CRAFT paper to further elaborate on this. In CRAFT [A], the authors take image sets of examples and use NMF to learn a “concept bank” and set of coefficients. The resultant concept bank is a learned set of basis vectors, and therefore each element does not have any semantic or conceptual significance on its own. To label each atom in this concept bank (C1, C2, …), the paper states that they must “be interpreted by looking at crops that maximize the NMF coefficient” and consider “new sets of images containing [the concept]” (see captions of Fig. 2, 4 of [A]). By feature visualization, we refer to this process of labeling atoms by visualizing either test set exemplars and crops (as done in CRAFT and Sparse Autoencoder analysis) or synthetic, generated images (as is common in mechanistic interpretability neuron analysis) that highly activate each atom and describing these sets of images qualitatively [B, C (section “Manual Human Analysis”)]. **This is a key difference between SpLiCE and CRAFT/NMF:** for CRAFT/NMF, describing or interpreting dictionary atoms requires a qualitative summary of the images that activate said atom. In contrast, SpLiCE’s dictionary atoms are automatically interpretable because they inherently correspond to text phrases, thus not requiring this qualitative analysis. For more discussion of how our method relates to NMF and other dictionary learning methods, we kindly point the reviewer to our discussion with reviewer djnP.
> We hope this cleared up any confusion our terminology may have caused.
>
> Thank you again for your engagement and support of the paper, and please let us know if you have any remaining questions!
>
> [A] Fel, T., Picard, A., Bethune, L., Boissin, T., Vigouroux, D., Colin, J., ... & Serre, T. (2023). Craft: Concept recursive activation factorization for explainability. In Proceedings of the IEEE/CVF Conference on Computer Vision and Pattern Recognition
>
> [B] Olah, C., Mordvintsev, A., & Schubert, L. (2017). Feature visualization. Distill
>
> [C] Bricken, T., Templeton, A., Batson, J., Chen, B., Jermyn, A., Conerly, T., ... & Olah, C. (2023). Towards monosemanticity: Decomposing language models with dictionary learning. Transformer Circuits Thread

---

### Author Rebuttal · Authors · 2024-08-07

We sincerely thank the reviewers for their thorough assessment of our paper and the AC for facilitating the discussion of our work. We appreciate the reviewers’ recognition that our paper is an “important study toward understanding the latent space of vision-language models like CLIP” and that it will “help to improve the next generation of such models in training and downstream use” [nCHL]. Furthermore, the reviewers note the paper “strives to address a vitally important task” [djnP], that “the research topic is worth pursuing” [jbkR], and that it “opens up the door to more intriguing work” [Vxod]. Finally, we are pleased to note that reviewers appreciated the novelty of the work [acbD, Vxod], the quality of the writing [all reviewers], and the clarity of the work [all reviewers]. In the following section we summarize the main points made by reviewers and respond to their comments. We also present additional experimental results, including results from a user study and evaluation on four additional benchmark datasets in the additional results section and in the body of our responses below.

**Human Evaluation.** Reviewers nCHL, acbD, and jbkR all asked for a human evaluation to validate our claims of improved interpretability. We present results from a small-scale user study in Fig. 1 of the additional results. We base our study off of that provided by reviewer acbD [23], benchmarking our method against two similar CLIP interpretability methods: LF-CBMs [23] and IP-OMP [22]. We provided users with twenty randomly chosen, correctly predicted images from ImageNet and two explanations comprising six concepts each for every image. We then asked users to evaluate the concept-based explanations for their relevance to the provided image inputs, their relevance to model predictions, and their informativeness. We find that users significantly preferred SpLiCE to the two baselines for relevance to the images and informativeness, with significance determined via a one-sample two-sided t-test and a threshold of p=0.01 (Additional material Figure 1). We also highlight that our method is able to produce similar/better concept decompositions, in terms of human interpretability, than the baselines without needing to train a classification probe or use class labels for concept mining, both of which are computationally expensive.

**Limitations of Concept Set.** Reviewers djnP, nCHL, abcD, and Vxod all mention the limitations of our LAION-based vocabulary, mainly related to its “task-agnosticity,” given that we construct this vocabulary using a specific dataset. While there certainly exist tasks that this vocabulary may fail at, it is intended to be a general purpose vocabulary that reflects the concepts learned by CLIP. We pick the LAION dataset because it is the training set of Open-CLIP (the model we use in this work) itself, but we acknowledge that it may still be a suboptimal dataset. Despite this, we find that this dataset performs well on a variety of tasks (CIFAR100, MIT States, WaterBirds, SUN397, Caltech 101, STL10, PASCAL VOC 2007). We agree with the reviewers that "task-agnostic" may be too strong, but it is accurate to state that our method is out-of-the-box applicable to a wide variety of tasks (owing to the generality and scale of LAION), and easily modified for others (i.e., simply listing additional concepts as opposed to labeling images and training a new model). Finally, we benchmark our vocabulary against the previous SOTA concept set, generated by GPT, in Figure 5, and explore other concept sets in Appendix C.10. We also note that our method can accommodate any user-defined vocabulary if the authors’ proposed one is insufficient for the application.  We will be sure to acknowledge these limitations in the final version of our paper and highlight that selecting a proper dictionary is an open question.

**Necessity of Theoretical Analysis.** Reviewers djnP, nCHL, and Vxod were concerned about the practicality of the assumptions in our theoretical analysis in Section 3 and their necessity to the core results and message of our paper. The intent of Section 3 is to reason from first principles regarding which mathematical properties enable vision-language models to have a sparse decomposition. To this end, we identified five assumptions sufficient to derive Proposition 1, i.e., the sparse decomposition property, similar to past work theoretically characterizing the linear representation hypothesis [11] and word2vec behavior [10]. We did not mean to claim that CLIP always satisfies all these assumptions, but rather that given our empirical results indicating our ability to apply a sparse decomposition to CLIP, we can reasonably conclude that CLIP may approximately satisfy assumptions 1-5.

We empirically explore the extent to which these assumptions hold in Appendix B.5. Reviewer djnP noted that these “critical” experiments would be much more useful in the main body of the paper as evidence of the practicality of our claims, further improving the paper’s clarity. Based on this feedback, we will move our empirical investigation to Section 3 and transfer the bulk of our theoretical analysis to the appendix, instead writing a shorter paragraph summarizing the basic assumptions and behaviors of CLIP required to admit a sparse decomposition.

---

### Author Response · Authors · 2024-08-12
**Discussion Summary**

We would like to thank the AC for facilitating the review of our paper and the reviewers for providing constructive feedback and engagement during the discussion period. In this work, we propose SpLiCE, a method that leverages sparse coding to interpret CLIP embeddings by representing them as sparse combinations of semantic concepts. Our interpretable CLIP embeddings maintain high performance and unlock novel use cases, including bias detection, data summarization, distribution shift monitoring, and more.

The reviewers' main concerns were with the lack of human evaluations, the limitations of our dictionary, and our theoretical analysis. We accordingly provided results from a small-scale user study following the suggested format of prior work, which highlighted the human interpretability of our method. We also further elaborated on the limitations of our proposed dictionary and discussed changes to our theoretical analysis in Section 3 of the paper. Following these rebuttals, reviewer abcD noted “most of [their] concerns were well addressed” and reviewer djnP “believe[s] it would be of great value to be presented at the conference”, and they increased their scores to a 6 and a 7 respectively. Reviewers nCHL, jbkR, and Vxod have not responded to our rebuttal as of yet.

We will be sure to incorporate the changes suggested by the reviewers into our camera-ready version. We also intend to expand our user study in this time, as suggested by reviewer acbD.

---

### Decision · Program_Chairs · 2024-09-25

**Decision:**

Accept (poster)

**Comment:**

Reviewers positively assessed the novelty and significance of providing interpretable embeddings by using dictionary learning to decompose CLIP representations.  Multiple reviewers also expressed the need to include human evaluations in the paper, which were provided by the authors during the rebuttal period.  Two reviewers were concerned about the paper's claims of providing increased interpretability at "no cost" to accuracy.  The authors agreed during the discussion to revise the abstract and several sentences in the paper to convey that the method can provide increased interpretability at a low cost to accuracy.  Given the strengths of the paper and the authors' willingness to make the corrections requested by reviewers, I am recommending acceptance of this paper.